# scCASE: accurate and interpretable enhancement for single-cell chromatin accessibility sequencing data

Songming Tang[1], Xuejian Cui[2], Rongxiang Wang[3], Sijie Li[1], Siyu Li[4], Xin Huang[5] & Shengquan Chen[1] ✉

Single-cell chromatin accessibility sequencing (scCAS) has emerged as a valuable tool for interrogating and elucidating epigenomic heterogeneity and gene regulation. However, scCAS data inherently suffers from limitations such as high sparsity and dimensionality, which pose significant challenges for downstream analyses. Although several methods are proposed to enhance scCAS data, there are still challenges and limitations that hinder the effectiveness of these methods. Here, we propose scCASE, a scCAS data enhancement method based on non-negative matrix factorization which incorporates an iteratively updating cell-to-cell similarity matrix. Through comprehensive experiments on multiple datasets, we demonstrate the advantages of scCASE over existing methods for scCAS data enhancement. The interpretable cell type-specific peaks identified by scCASE can provide valuable biological insights into cell subpopulations. Moreover, to leverage the large compendia of available omics data as a reference, we further expand scCASE to scCASER, which enables the incorporation of external reference data to improve enhancement performance.

The emergence of single-cell omics sequencing technology allows us to study cellular heterogeneity and complexity[1]. In the field of single-cell epigenomics, single-cell chromatin accessibility sequencing (scCAS) can measure the open chromatin status of individual cells[2], discover epigenomic heterogeneity, reveal gene regulation mechanisms, and deepen our understanding of life processes such as cell development, cell differentiation, and diseases[3–6]. However, compared with conventional single-cell RNA sequencing (scRNA-seq) data, scCAS data present some challenges due to assay-specific characteristics, such as high dimensionality and severe sparsity[7]. In addition, though there are many thousands of possible open positions per cell, only a few thousand distinct reads can be captured, resulting in the dropout events of sequencing data[2,8–10]. The dropout events of scCAS data significantly impact downstream analyses such as cell clustering and data visualization[10]. Consequently, developing an efficient scCAS data enhancement method to impute the dropout events is essential for not only improving the quality of downstream analyses but also revealing biological insights.

Many enhancement methods have been designed for scRNA-seq data, such as MAGIC[11], scImpute[12], DCA[13], and SAVER[14], and have also been attempted to be applied to scCAS data[15]. However, since these methods are not explicitly designed for scCAS data, they exhibit poor performance, lack the necessary level of stability when applied to scCAS data, and should be used cautiously[10,15], highlighting the pressing need for enhancement methods specific to scCAS data. Currently, several computational methods have been proposed to enhance scCAS data. For instance, SCALE embeds each cell with a vector of latent features via an encoder network and reconstructs original

[1]School of Mathematical Sciences and LPMC, Nankai University, Tianjin 300071, China. [2]MOE Key Laboratory of Bioinformatics and Bioinformatics Division of BNRIST, Department of Automation, Tsinghua University, 100084 Beijing, China. [3]Department of Computer Science, University of Virginia, Charlottesville, VA 22903, USA. [4]School of Statistics and Data Science, Nankai University, Tianjin 300071, China. [5]Beijing Key Laboratory for Radiobiology, Department of Radiation Biology, Beijing Institute of Radiation Medicine, 100850 Beijing, China. ✉e-mail: chenshengquan@nankai.edu.cn

profiles through a decoder network to obtain the enhanced open chromatin status of each cell[9]. scBFA models the gene detection patterns through binary factor analysis to obtain an embedding matrix and a loading matrix that can be multiplied to enhance both scRNA-seq and scCAS data[16]. scOpen uses regularized non-negative matrix factorization (NMF) on scCAS count matrix and restores the original data using the factorized matrices to enhance and denoise the scCAS data[10]. Besides, scBasset adopts a deep convolutional neural network to leverage DNA sequence information underlying accessibility peaks to model scCAS data and predict chromatin accessibility[17].

However, certain limitations deserve further consideration. On the one hand, existing methods primarily focus on enhancing scCAS data only on accessibility peaks, disregarding the crucial factors of cell-to-cell difference and correlation, which can be used to enhance epigenetic signals and maintain cellular heterogeneity within and between subpopulations[15,18]. On the other hand, large compendia of available omics data offer valuable prior knowledge that can be employed to model the target scCAS data preferably. Studies have consistently demonstrated that leveraging publicly available data as references can be highly effective in aiding single-cell omics data analysis[19–21]. Despite that, none of the existing scCAS data enhancement methods can effectively incorporate external reference data.

To fill these gaps, we propose an accurate and interpretable scCAS data enhancement method, namely scCASE, based on non-negative matrix factorization. To model the cell-to-cell difference and correlation, we incorporate a cell-to-cell similarity matrix when performing non-negative matrix factorization, which can be used to aggregate the scCAS data of similar cells and be updated in an iterative manner. The enhanced scCAS data effectively capture signals of cellular heterogeneity and improve the quality of downstream analyses, such as cell clustering and data visualization. With comprehensive experiments on one simulated and ten publicly available real scCAS datasets, we demonstrated that scCASE outperforms baseline methods in the enhancement of scCAS data. Moreover, we showed that we can identify cell type-specific peaks based on the trained scCASE model and unveil biological insights into specific cell subpopulations via extensive biological function enrichment, tissue-specific expression enrichment, and partitioned heritability analysis, suggesting the

compelling interpretability of scCASE. Furthermore, we developed scCASER, which can incorporate publicly available omics data as reference data to better characterize the target scCAS data and facilitate data enhancement. We also provided various approaches to construct effective reference data, further expanding the applicability of scCASER.

## Results

### The scCASE model

The schematic diagram of scCASE is shown in Fig. 1. The input of scCASE is a scCAS count matrix, which is processed to filter the peaks accessible in fewer than 1% cells and then subjected to a term frequency-inverse document frequency (TF-IDF) transformation ("Methods"). Based on non-negative matrix factorization (NMF), scCASE incorporates an iterative updated cell-to-cell similarity matrix to enhance scCAS data. Given that similar cells generally have similar chromatin accessibility patterns, we can describe the read counts of a certain cell as a weighted average of other cells by introducing the similarity matrix to fully utilize cell-to-cell correlation. The higher the similarity between a cell and the certain cell, the greater the weight of the cell. We also generated a matrix randomly sampled by binomial distribution to prevent similar cells from exhibiting entire consistent read counts. The model can be divided into two parts. On the one hand, via multiplying the scCAS count matrix by the Hadamard product of the similarity matrix and the randomly sampled matrix, we can obtain a scCAS count matrix enhanced by similarity. On the other hand, we can obtain a reconstructed scCAS count matrix via multiplying the two factorized matrices of projection matrix and cell embedding matrix. We iteratively update the similarity matrix, the projection matrix, and the cell embedding matrix to minimize the difference between the enhanced matrix and the reconstructed matrix. In addition, scCASE can adaptively adjust the dimension of cell embedding based on the input dataset ("Methods").

We compared the performance of scCASE with state-of-the-art scCAS data enhancement methods, including SCALE[9], scBFA[16], scOpen[10], and scBasset[17] ("Methods"), using their default parameters, and conducted comprehensive comparisons on various datasets, including a simulated scCAS dataset, eight publicly available scCAS

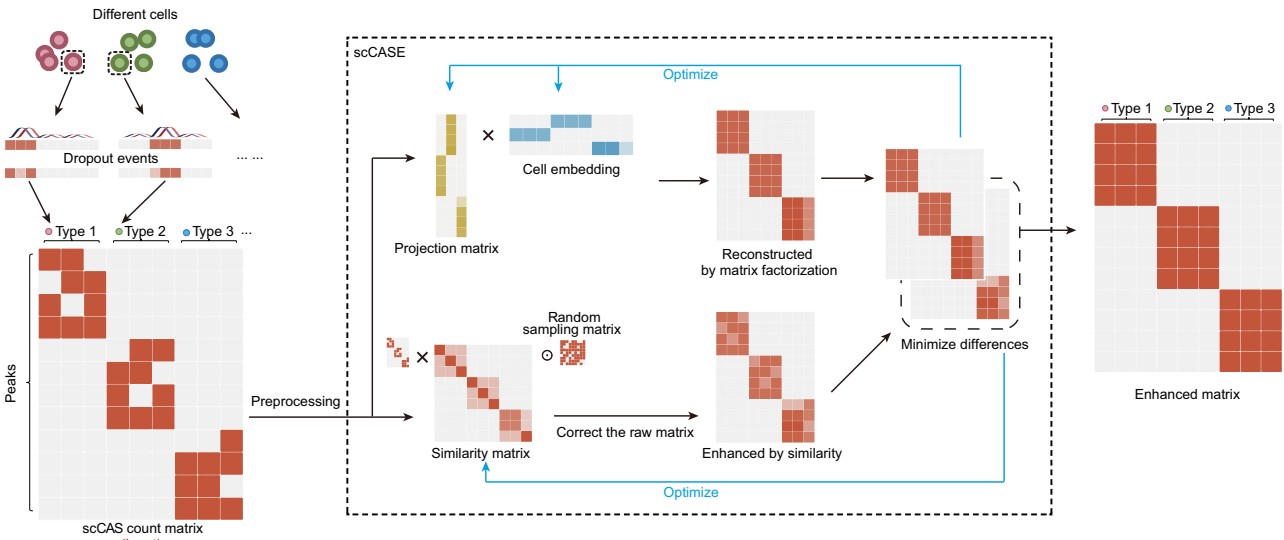

**Fig. 1 | The graphical illustration of scCASE.** scCASE takes a preprocessed scCAS count matrix as input, then generates an initial similarity matrix based on the matrix and performs non-negative matrix factorization to obtain an initial projection matrix and an initial cell embedding. We describe the read counts of a certain cell as a weighted average of other cells by introducing the similarity matrix to fully utilize cell-to-cell correlation. Random sampling matrix is generated through binomial distribution and Hadamard multiplied with similarity matrix in the computation is to avoid same cells exhibit almost the same accessible peaks which improperly reduces the cellular heterogeneity. The model uses similarity and matrix factorization to enhance scCAS data separately and iteratively optimizes the initialized matrix, aiming to minimize the difference between the reconstructed and enhanced matrices.

datasets, two scCAS datasets annotated base on the paired scRNA-seq data, and two mixed scCAS datasets ("Methods", Supplementary Table S1). The simulated data is constructed by simCAS[22], the state-of-the-art method for scCAS data simulation. For the eight publicly available scCAS datasets, the Blood dataset contains ten types of human hematopoietic cells from bone marrow, while the BM0828 dataset is a widely-used benchmarking subset of the Blood dataset with the donor label BM0828[7,21,23,24]. To determine whether the findings remain consistent across various species and tissues, we collected the WholeBrainA, WholeBrainB, LungA, LungB, Spleen, and LargeIntestine datasets, which were derived from the whole brain, lung, spleen, and large intestine tissues of adult mice[25]. We also employed two multiome datasets, including Muto and PBMC, for consistency between scRNA-seq and scCAS to reduce the impact of mislabeling caused by annotating solely based on scCAS data. Moreover, two mixed datasets are concatenated with datasets from different tissues or protocols. All the datasets vary in terms of protocol, species, tissue, number of cells, number of peaks, number of cell types, and imbalance degree of cell types, but all exhibit high sparsity and dimension (Supplementary Table S1).

## scCASE enhances scCAS data for better cellular heterogeneity characterization

We first assess the capability of scCASE in recovering dropout events. For a real dataset, the ground truth without any corruption is unknown, we thus quantified data enhancement performance on a simulated dataset. We utilized simCAS[22], the state-of-the-art method for scCAS data simulation, in discrete mode to construct a dataset consisting of five clusters, each containing 500 cells. We then used the EpiScanpy pipeline to identify the top 3000 differentially accessible peaks for each cell type[26]. In this way, we obtained a simulated dataset with 2500 cells and 15,000 peaks as the input of different enhancement methods (Fig. 2a and Supplementary Table S1). Figure 2b shows the heatmap of the data enhanced by scCASE, which effectively filled in the dropout events in the raw data (Supplementary Fig. S1a) and provided clearer cell type-specific patterns. Next, we evaluated the correlation between enhanced simulated data and ground truth data with the area under the precision-recall curve (auPRC) and the area under the receiver operating characteristic curve (auROC) for each cell and each peak, respectively. scCASE significantly outperformed the competing methods by presenting higher auPRC and auROC regardless of cell-wise (Fig. 2c) or peak-wise (Supplementary Fig. S1b, c). In addition, we calculated the auPRC and auROC for each of the five cell types separately (Supplementary Fig. S1d, e) and observed that even for cell types with significant dropout events, such as cell types A and D, scCASE was able to uncover the potential epigenetic information and accurately enhance the scCAS data.

Furthermore, we evaluated the ability of scCASE to accurately capture the underlying characteristics of cells and effectively promote cell clustering performance in real datasets. Utilizing the same evaluation strategy as scOpen, a method tailored for scCAS data enhancement, we performed principal component analysis (PCA) to reduce the dimensionality of the enhanced data to 50 dimensions[10] and then performed Louvain clustering with binary search strategy to ensure the number of clusters equals to the number of cell types[7,26,27]. The clustering results were then evaluated by adjusted Rand index (ARI)[28], adjusted mutual information (AMI)[29], and Fowlkes-Mallows index (FMI)[30]. Besides, following scOpen, we examined the influence of data enhancement on the estimation distances between cells by silhouette score[10,31]. A higher silhouette score indicates that a cell exhibits more significant similarity to cells of the same type compared to that of other types ("Methods").

Compared with baseline methods, scCASE demonstrated the overall best performance (Fig. 2d, e; Supplementary Fig. S2 and Supplementary Table S2). Specifically, scCASE achieved an average

improvement of 13.89% in ARI and 24.41% in silhouette scores across the eight datasets compared with the second-best method for each dataset (Supplementary Fig. S3). Besides, the scCAS data enhanced by scCASE significantly outperformed the raw data and the data enhanced by other methods with $p$ values less than 0.05 in one-sided paired Wilcoxon signed-rank tests (Supplementary Fig. S4), indicating that scCASE can effectively enhance scCAS data and thus characterize cellular heterogeneity well and improve the accuracy of cell clustering. Annotation solely based on scCAS data may lead to mislabeling, thus, these labels aren't necessarily better at capturing true biological heterogeneity. We included the two scCAS datasets annotated based on paired scRNA-seq data including Muto and PBMC in our benchmark ("Methods"), and the data enhanced by scCASE consistently demonstrated the best performance compared to raw data and data enhanced by baseline methods (Supplementary Fig. S5). Moreover, scCASE can obtain latent cell embeddings, achieving data dimensionality reduction. The cell embeddings from scCASE are also beneficial for clustering. The embeddings learned by scCASE demonstrate certain advantages over the latent representations obtained from baseline methods (Supplementary Fig. S6).

To visually illustrate that the real scCAS data can be efficaciously enhanced by scCASE, we further provided an example using the BM0828 dataset. We utilized EpiScanpy to identify the top ten differentially accessible peaks for each type[26]. Based on the differentially accessible peaks identified from raw data, we plotted heatmaps of the count matrices of the raw data (Fig. 2f) and the data enhanced by scCASE (Fig. 2g). Evidently, the raw data exhibited significant dropout events and technical noise, and showed indistinct patterns of different cell types. After enhancement by scCASE, the dropout events have been effectively imputed, and accessible patterns of different cell types can be effectively captured, again suggesting the profound proficiency of scCASE in interrogating and elucidating cellular heterogeneity. Moreover, correcting for sequencing depth is one of the most important goals in the enhancement of scCAS data, and many commonly used methods for preprocessing scCAS data such as TF-IDF don't fully correct sequencing depth[25]. Following Cusanovich et al.[25], we applied TF-IDF transformation to the raw count matrix and employed SVD to reduce the model dimensions to 10, a significant correlation between principal component 1 (PC1) and sequencing depth can be observed. The correlation coefficient between PC1 and sequencing depth exceeds 0.8 in the Blood dataset, and 0.7 in the LungA dataset (Supplementary Figs. S7 and S8). This is also evident in uniform manifold approximation and projection (UMAP) visualization, where sequencing depth largely determines the positions of cells in the low-dimensional representation (Fig. 2h and Supplementary Figs. S7a and S8a). For the data enhanced by scCASE, the observed correlation was attenuated. In the Blood dataset, this correlation with PC1 has been reduced by 45.6%, while in the LungA dataset, it has been reduced by 19.6% (Fig. 2i and Supplementary Figs. S7 and S8). To enhance the capability of scCASE in further mitigating the impact of sequencing depth, we extended the scCASE model and set it as an optional variant (Supplementary Text S3). In the data enhanced by the extended scCASE, it is hard to observe a strong correlation between individual principal components and the sequencing depth (Supplementary Figs. S7 and S8).

Moreover, we visualized the enhanced scCAS data using t-SNE (t-distributed stochastic neighbor embedding) and UMAP. Compared with the raw data and the data enhanced by other methods, utilizing the data enhanced by scCASE, we can better separate different cell types and recognize subtle differences between cells of different types (Fig. 3 and Supplementary Figs. S9 and S10). For example, we can effectively distinguish the LMPP, MEP, and CMP in the Blood and BM0828 datasets using the data enhanced by scCASE (Fig. 3a, b). In contrast, these cell types could not be separated effectively in the raw data and the data enhanced by SCALE and scBFA. Besides, in the LungA

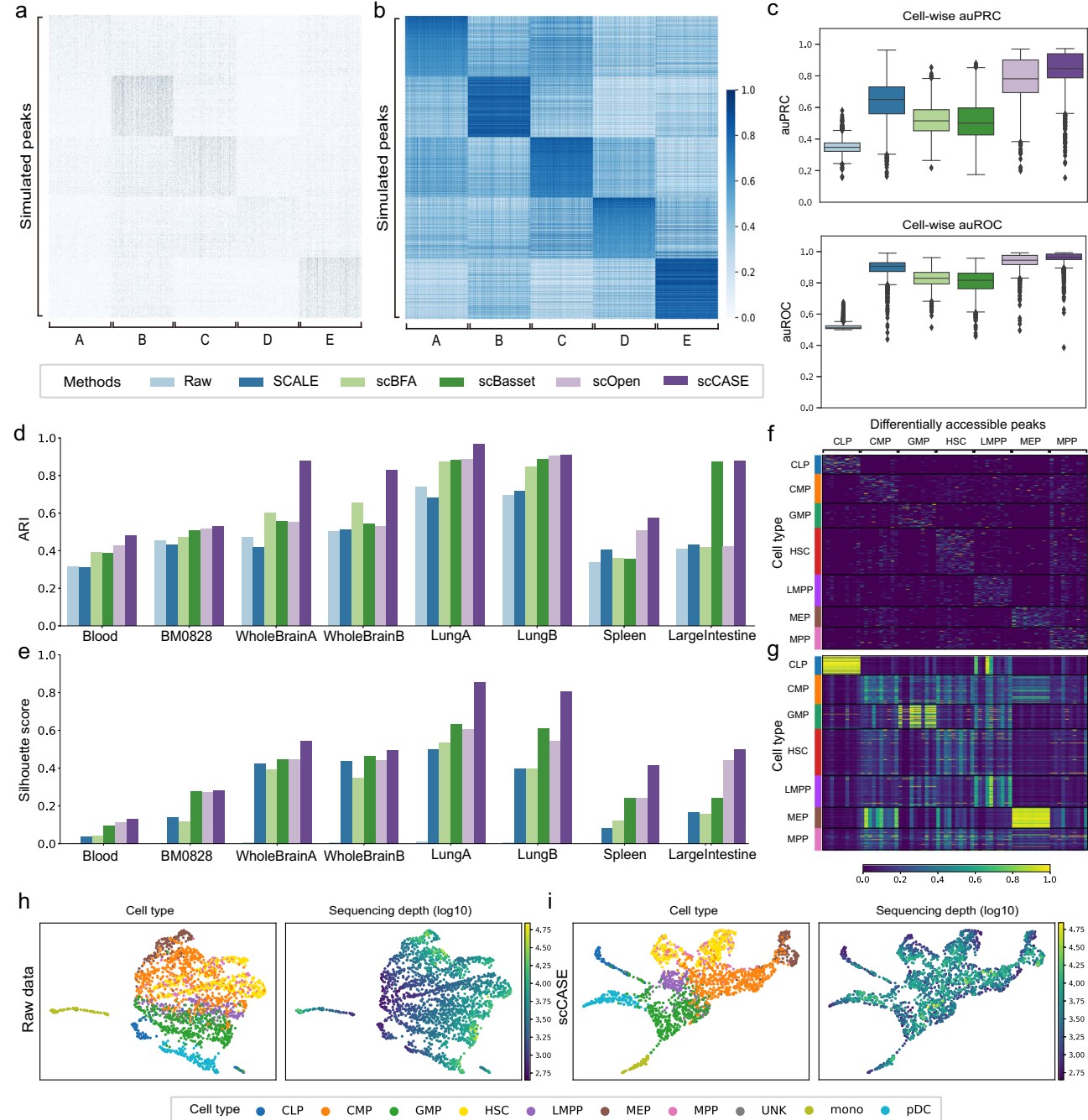

**Fig. 2 | Performance of different scCAS data enhancement methods. a** Heatmap of the simulated scCAS count matrix. The *y*-axis represents different peaks while the *x*-axis represents different cells. **b** Heatmap of the matrix enhanced by scCASE. **c** The boxplot of the cell-wise auPRC and auROC between the ground-truth simulated data and the data enhanced by different methods (*n* = 2500 cells). The midline represents the median, the boxes represent the interquartile range, whiskers represent 1.5× interquartile and points represent outliers. **d** The clustering performance assessed by ARI scores on datasets enhanced by different methods. **e** Silhouette scores according to cell type labels on datasets enhanced by different

methods. **f** Heatmap of the count matrix of raw data for the differentially accessible peaks identified from raw data on the BM0828 dataset. **g** Heatmap of the matrix enhanced by scCASE for the differentially accessible peaks identified from raw data on the BM0828 dataset. **h** UMAP visualization of the raw Blood dataset. Cell type labels and sequencing depth (log10) are projected onto the visualizations, respectively. **i** UMAP visualization of the scCASE-enhanced Blood dataset. Cell type labels and sequencing depth (log10) are projected onto the visualizations, respectively. Source data are provided as a Source Data file.

and LungB datasets, alveolar macrophages, dendritic cells, and some B cells were close to each other in the latent space of raw data (Fig. 3c, d). Using the data enhanced by other methods, we struggled to distinguish these types ideally, yet only scCASE proved capable of characterizing the cellular heterogeneity of these cell types. Furthermore, on the Spleen dataset, we can identify T cells and regulatory T cells using the data enhanced by scCASE, while they cannot be identified via

the data enhanced by other methods (Fig. 3e), indicating that even for closely related cell types, scCASE can still capture the subtle differences between different subtypes.

### scCASE intuitively reveals cell type-specific biological insights
scCASE can extract latent features of cell subpopulations, which can be applied to downstream analyses and reveal cell type-specific biological

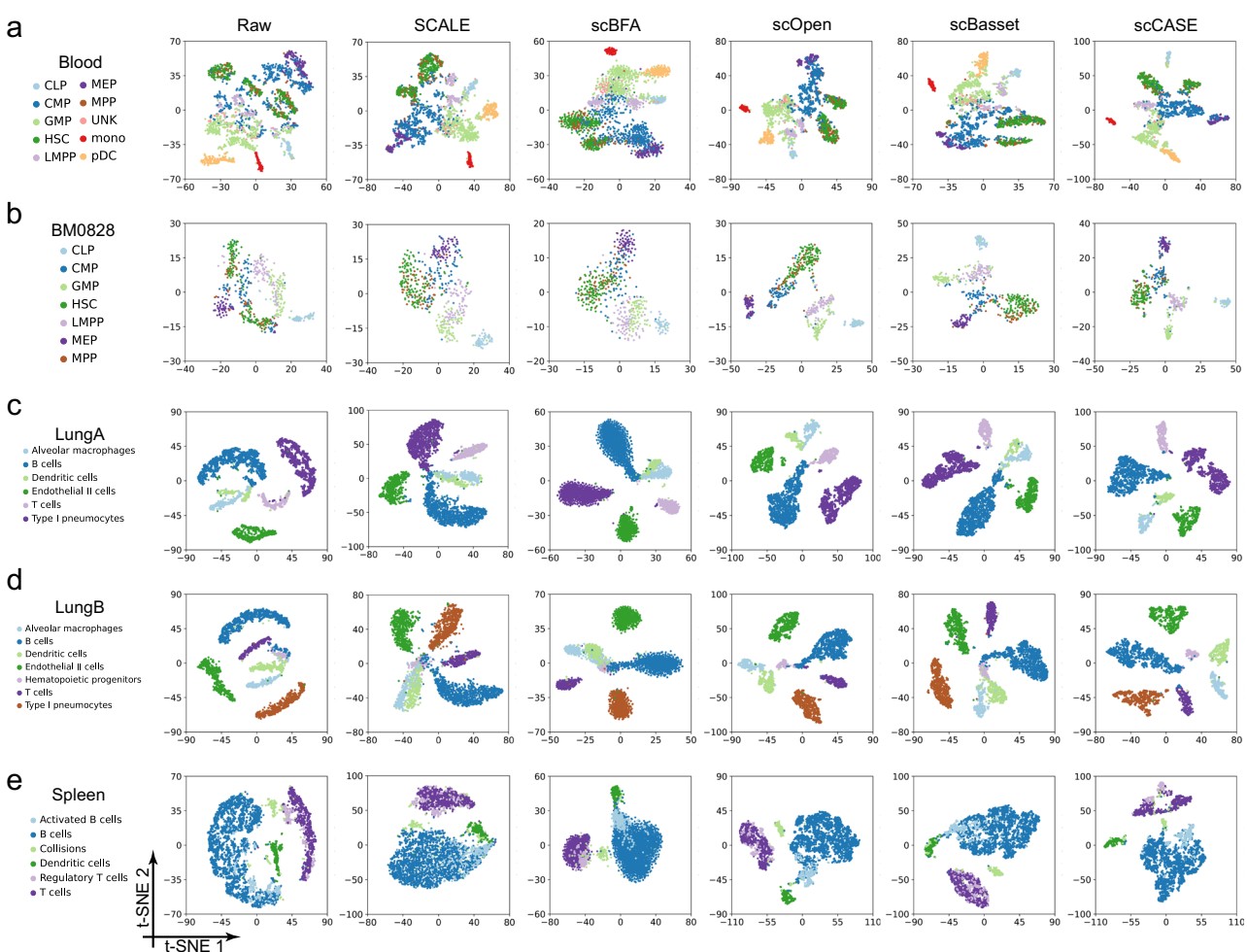

**Fig. 3 | t-SNE visualization of the raw scCAS data and the data enhanced by different methods. a** The Blood dataset. **b** The dataset of BM0828. **c** The LungA dataset. **d** The LungB dataset. **e** The Spleen dataset. Source data are provided as a Source Data file.

insights. Specifically, scCASE can identify cell type-specific peaks, allowing us to gain insights into various aspects of cellular heterogeneity, such as functionality, tissue-specific expression, and partitioned heritability. Taking the Blood dataset as an example, we utilized the factorized cell embedding and projection matrices to identify the specific accessibility peaks of monocytes. In Fig. 4a, each row represents a component of cell embedding learned by scCASE, and each column represents a cell, with cells ordered by cell type. We considered a column of the projection matrix (a pattern of peaks) corresponding to the row of cell embedding with the highest activation levels in the monocyte cluster, investigated the pattern of peaks with relatively large coefficients, and identified the top 100 influential peaks as the monocyte-specific peaks (Fig. 4a). Subsequently, we performed a genomic region enrichment of annotation tool (GREAT)[32] analysis on the monocyte-specific peaks identified by scCASE (Supplementary Table S3). The top five pathways with the smallest $p$ values using the binomial test consist of response to bacterium; response to lipopolysaccharide; response to molecule of bacterial origin; positive regulation of immune system process; and regulation of immune system process. All the enriched top five pathways are consistent with the known functions of monocytes: monocyte is a subgroup of white blood cells, an important part of the defense system, and an essential natural immune effector cell.

To better demonstrate the correlation between cell type-specific peaks and functionality, we further used the UCSC genome browser[33] and the UniProt database[34] to retrieve these peaks. Some monocyte-specific peaks are located within or close to the genes' transcript positions expressed explicitly in monocytes, indicating that scCASE can be used to identify genomic regions highly related to cell functions. For example, the identified peak chr2: 219,246,758-219,247,258 (hg19) is located within the genomic region of the human *SLC11A1* gene. This gene encodes a natural resistance-associated macrophage protein that transports divalent transition metals, and it is highly expressed in macrophages that differentiate from monocytes[35]. The peak chr6: 41,239,630-41,240,130 (hg19) is located near the genomic region of the *TREM1* gene, which is selectively expressed in subpopulations of monocytes in the blood and mediates the activation of monocytes[36].

To investigate whether these strategies of scCASE for revealing cell type-specific biological insights can be applied to datasets from different species and tissues, we conducted the same experiments on the LungB dataset and took T cells as an example. The top five pathways of peaks identified by scCASE with the smallest $p$ values of the binomial test in GREAT analysis are abnormal T cell physiology; abnormal CD8-positive, alpha-beta T cell morphology; abnormal T cell activation; abnormal CD8-positive, alpha-beta T cell number; and abnormal T cell proliferation (Supplementary Fig. S11 and Supplementary Table S4), which have good agreement with the functionality of T cells. In addition, the scCASE-identified peak chr12: 114,092,788-114,093,703 (mm9) is contained in the genomic regions of the *Gpr132* gene, which serves as a mechanism to slow down proliferation and repair damaged DNA in T and B lymphocytes (Supplementary

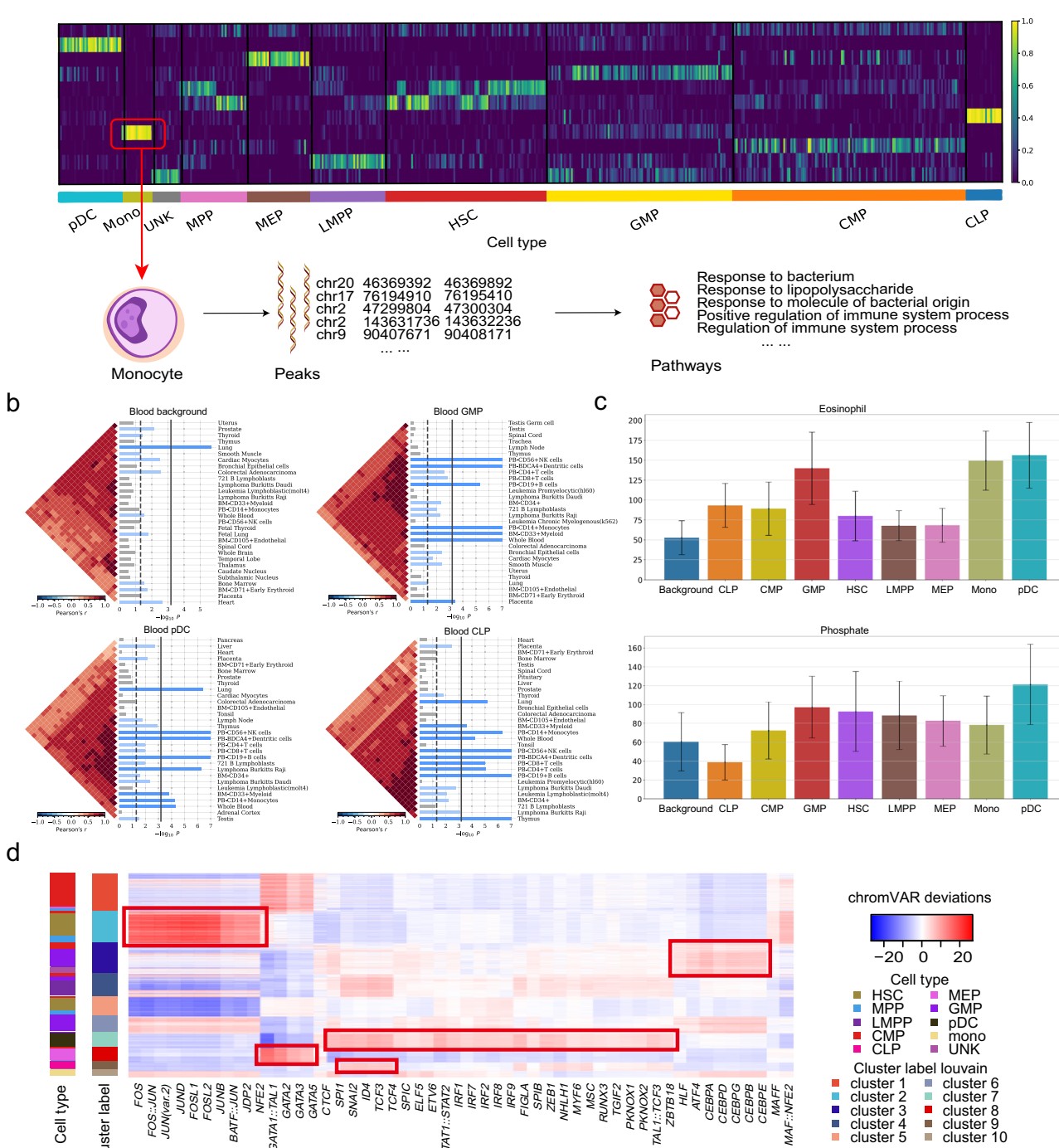

**Fig. 4 | scCASE can effectively reveal cell type-specific biological insights.**
**a** Heatmap of the cell embeddings in latent space. Each row represents a component of cell embedding learned by scCASE, and each column represents a cell, with cells ordered by cell type. **b** Cell type-specific peaks and background peaks were subjected to SNPsea analysis to determine the top 30 significantly enriched tissues. Bars represent the empirical *p* values calculated by the SNPsea algorithm. The significance of gene enrichment for a specific annotation was assessed using one-sided *p* value cutoffs at the 0.05 level, represented by vertical dashed and solid lines. The unadjusted *p* value cutoff was used for the dashed line, while the Bonferroni-corrected *p* value cutoff was applied for the solid line. The heatmaps

display the Pearson correlation coefficients, which measure the similarity between expression profiles. The expression profiles were ordered using hierarchical clustering with the unweighted pair-group method with arithmetic means (UPGMA). **c** Heritability enrichment estimated by stratified LDSC of the SNPs within the cell type-specific peaks and the background peaks, for two blood-related traits. The error bars and centers of error bars represent the standard errors and average values of 200 equally sized jackknife blocks of adjacent SNPs about the estimates of enrichment. **d** The top 50 most variable TF binding motifs within the cluster-specific peaks for the cells in the Blood dataset. The deviations calculated by chromVAR are shown. Source data are provided as a Source Data file.

Table S5)[37]. Besides, the scCASE-identified peaks chr11: 46,201,573-46,203,658 and chr11: 46,157,147-46,158,328 are located within the genomic regions of the *Itk* gene, which is associated with T cell antigen receptor signal transduction and contributes to T cell activation[38]. The results suggest that the capability of scCASE to unveil cell type-specific biological insights can be generalized to various species and tissues.

We further demonstrated that the cell type-specific peaks identified by scCASE can also provide tissue-specific expression enrichment. We again used scCASE to identify 1000 cell type-specific peaks for each cell type to verify whether the peaks provide more significantly expressed tissue specificity than the background peaks without cell type specificity. We calculated the mean coefficient of the projection matrix and the mean coefficients of each peak. Next, we chose the top 1000 peaks whose mean coefficients are closest to the mean coefficient of the projection matrix as background peaks. Subsequently, we performed SNPsea analysis[39], using the default settings, on each set of cell type-specific peaks and background peaks separately to obtain tissues explicitly affected by these peaks ("Methods", Fig. 4b and Supplementary Fig. S12). For the Blood dataset, tissues associated with blood display significant enrichment in gene expression based on the identified cell type-specific peaks. However, less enrichment is observed based on the background peaks, indicating that the cell type-specific peaks identified by scCASE can exhibit distinct tissue specificity and thus provide cellular heterogeneity insights in related tissues.

Moreover, the cell type-specific peaks obtained by scCASE can contribute to investigating phenotype variations. We employed partitioned linkage disequilibrium score regression (LDSC)[40] with default settings to quantify heritability enrichment for phenotypes within cell type-specific and background peaks in the Blood dataset ("Methods"). Compared with the background peaks, we revealed a significant enrichment of heritability for blood-related phenotypes, specifically within the cell type-specific peaks (Fig. 4c and Supplementary Fig. S13).

Concretely, the enrichments of heritability for the eosinophil and the phosphate count in the cell type-specific peaks are higher than that in the background peaks except CLP cells. Therefore, scCASE possesses the potential to enhance comprehension regarding the significance of distinct cell subpopulations and establish systematic connections between particular phenotypes and the function of specific cell types.

Given that motif enrichment analysis can deepen the understanding of regulatory mechanisms, we also showed that scCAS data enhanced by scCASE can trenchantly reveal cell type-specific motifs. We performed PCA on the data enhanced by scCASE, followed by the Louvain clustering to obtain the cluster labels. Following RA3[7], we identified 1000 specific peaks for each cluster using the hypothesis testing procedure in scABC based on the scCASE- enhanced matrix[41]. Then, chromVAR[42] was utilized to determine the transcription factor (TF) binding motifs enriched within these cluster-specific peaks ("Methods"). We visualized the top 50 most variable TF binding motifs in the Blood dataset (Fig. 4d and Supplementary Fig. S14). Existing literature confirms that 35 motifs of the top 50 most variable TF binding motifs are associated with hemocytes (Supplementary Table S6). Some TF binding motifs are specific to one or two cell types, and existing literature further confirms the correlation between these TFs/motifs and the corresponding cell types. For example, the *BATF* gene, expressed explicitly in HSCs (hematopoietic stem cells), limits the self-renewal and differentiation checkpoint, and the *JUNB* gene regulates HSC function[43]. In MEP cells (megakaryocyte-erythroid progenitor cells), the specifically expressed motifs include the *GATA* family (*GATA1, GATA2, GATA3, GATA5*) and *TAL1*, of which the former involved in the regulation of the development of erythroid and megakaryocyte precursors and crucial for normal hematopoiesis, while the latter serving as a positive regulator of the erythroid lineage differentiation[44,45]. The pDCs (plasmacytoid dendritic cells) participate in immune function. The genes *SPI1, IRF7, IRF8, SPIC*, and *SPIB* are involved in immune function and necessary for developing plasmacytoid dendritic cells[46–49]. Moreover, the *CEBPA, CEBPB, CEBPD, CEBPE*, and *CEBPG* genes are expressed explicitly in GMP cells (granulocyte-macrophage progenitor cells) and are critical in normal granulocyte production and essential for the transitioning progress from CMP cells (common myeloid progenitor cells) to GMP cells[50–53].

Moreover, the enrichment results with the cell type-specific peaks identified by scCASE are relatively better than those identified by EpiScanpy (Supplementary Text S4 and Supplementary Figs. S15–S20), demonstrating the superior biological significance of the cell type-specific peaks identified by scCASE. Taken together, we can use scCASE to explore more comprehensive biological implications and gain broader cell type-specific insights.

## scCASE exhibits superior robustness to various application scenarios

We first assess the robustness of scCASE to different sparsities, imbalance degrees, and sample sizes of datasets. We employed the LungA dataset as an example given its high sparsity. We intentionally manipulated the data to simulate different conditions, allowing us to evaluate the robustness of different scCAS data enhancement methods under such extreme scenarios. We excluded the scBFA and scBasset methods from comparison in this section due to their inadequate speed and memory usage performance, rendering them incapable on the LungA dataset within 48 h under a memory limitation of 256GB. We randomly dropped out the non-zero entries to be translated to zero with a probability equal to the dropout rate, which was set to range from 0% to 80%. As the dropout rate increased, ARI of the clustering results obtained from the scCAS data enhanced by scCASE was stable and satisfactory, while the raw data and the data enhanced by other methods exhibited varying degrees of fluctuation and decline (Fig. 5a). Then, considering that the proportions of different cell types can impact the capability of methods to learn distinctive features of each type[24], we conducted random subsampling by reducing the differences in cell number of various cell types to assess the performance of different methods on datasets with various imbalanced degrees of cell types ("Methods"). The results revealed that scCASE consistently and accurately enhanced dropout events across different degrees of cell type imbalance (Fig. 5b), suggesting that scCASE is robust and proficient in handling datasets with imbalanced cell types. Next, we conducted a comprehensive validation to confirm the robustness of scCASE to sample size. We randomly subsampled cells in the raw data to decrease the number of cells from a total of 3671 cells to 367 (about one-tenth of the total). As shown in Fig. 5c, regardless of the dataset size, scCASE consistently achieved superior and stable enhancement performance for cellular heterogeneity characterization.

Secondly, with the accumulation of public scCAS data, datasets from different sources, tissues, donors, or batches pose great challenges for data analysis[54]. We concatenated two mouse brain datasets from protocols of 10X and snATAC[55,56] as a new dataset named Mixed-protocols, which exhibits obvious batch effects. Besides, to evaluate the performance of scCASE in datasets affected by different tissues, we concatenated the LungA and Spleen datasets[25] as a new dataset named Mixed-tissues (Supplementary Table S1). We evaluated the clustering performance of data enhanced by different methods and found that scCASE consistently outperformed the baseline methods, indicating that scCASE can also effectively enhance data from different sources, tissues, donors, or batches for better downstream analysis (Fig. 5d and Supplementary Figs. S21 and S22). Moreover, we also provide an optional extension of scCASE, enabling it to significantly enhance the analyses of data with batch effects (Fig. 5e), and this modification will be provided as an optional variant (Supplementary Text S5). Taking the Mixed-protocols dataset as an example, this dataset exhibits obvious batch effects. The extended scCASE achieved a significant lead in

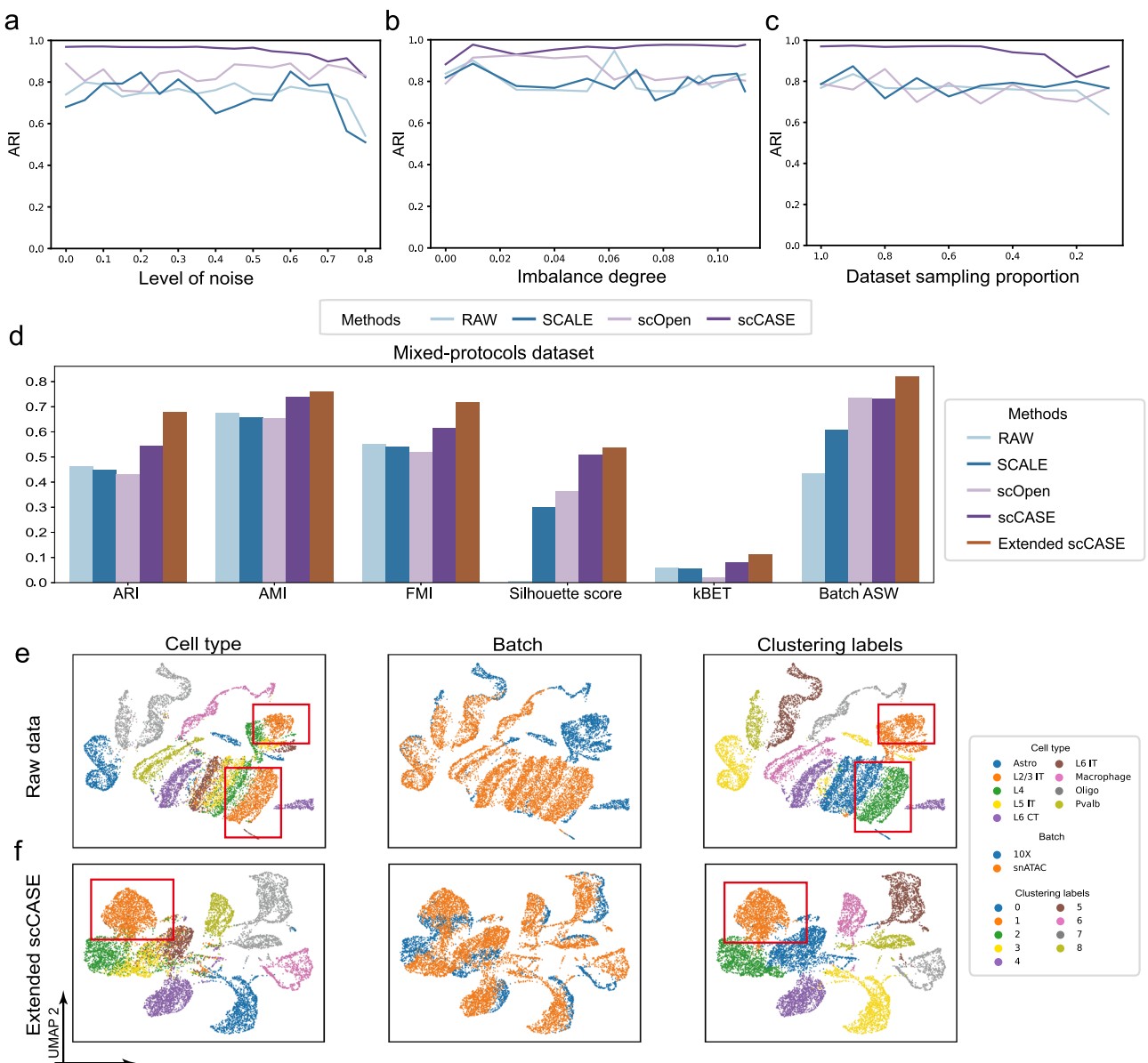

**Fig. 5 | Robustness tests and batch effect correction. a–c** Robustness of different methods in various scenarios. **a** Robustness of different methods to the noise level. **b** Robustness of different methods to the cell type imbalance degrees. **c** Robustness of different methods to the dataset size. **d** The values of different metrics of raw data and the data enhanced by various methods on the Mixed-protocols dataset. UMAP visualizations of (**e**) raw data and (**f**) the data enhanced by scCASE on the Mixed-protocols dataset. Cell type annotation labels, batch labels and clustering labels are projected onto the visualizations. Source data are provided as a Source Data file.

metrics measuring both the preservation of biological variation (ARI, AMI, FMI, and Silhouette score), and batch mixing (k-nearest neighbor batch effect test (kBET) and modified average silhouette width of batch (Batch ASW)) (Supplementary Text S1 and Supplementary Fig. S21a)[57,58]. In the raw data and data enhanced by baseline methods, L2/3 IT cells from different baches exhibit significant differences, making it challenging to identify them as the same cell type in UMAP visualization and Louvain clustering (Fig. 5e and Supplementary Fig. S21b–e). The extended scCASE is able to reduce the differences between L2/3 IT cells from different batches, resulting in their clustering as a single group in Louvain clustering (Fig. 5f and Supplementary Fig. S21f). The results indicate that the extended scCASE can effectively correct batch effects and facilitate the analyses of data with batch effects.

Taken together, scCASE can enhance scCAS data effectively under diverse scenarios, demonstrating its strong robustness and suggesting

that it can be widely employed to facilitate data enhancement on scCAS datasets with different characteristics.

## scCASE can effectively incorporate reference data to further promote performance

Incorporating the available omics data in public databases as a reference for the analysis of single-cell omic data is a practical approach to address the inherent challenges posed by high levels of noise and technical variation[7,20,21]. However, current scCAS data enhancement methods rely on the target scCAS data itself, neglecting the valuable prior information contained within the extensive public chromatin accessibility data. Given that reference data offer a more comprehensive representation of the general chromatin accessibility landscape for a specific cell type, we have expanded scCASE method and developed scCASER (scCASE with reference data) to further utilize the prior information (Fig. 6a). Briefly, scCASER uses the reference data as

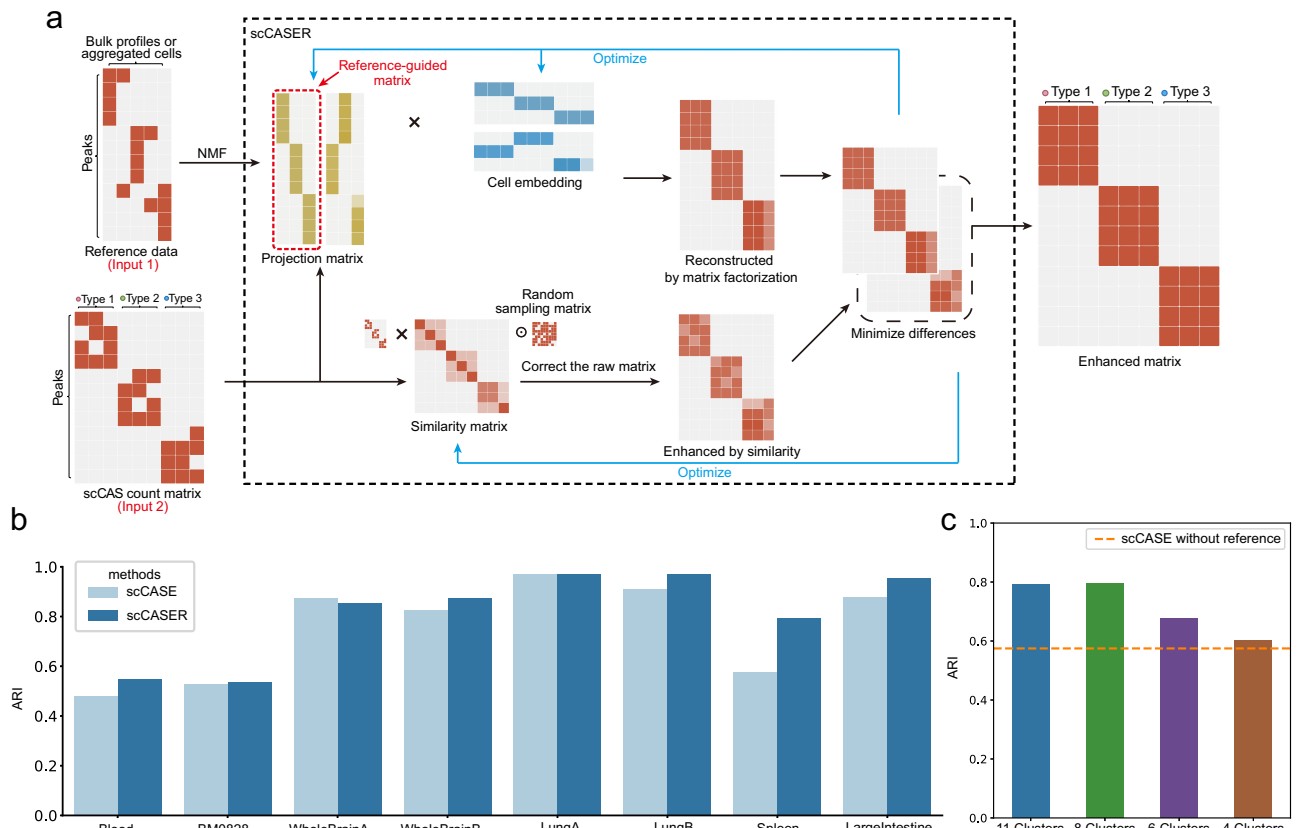

**Fig. 6 | scCASE can effectively incorporate reference data to further promote enhancement performance. a** The graphical illustration of scCASER. scCASER builds upon scCASE by additionally incorporating reference data, which serves as the second input to the model. scCASER performs NMF on the reference data and incorporates the factorized projection matrix from the reference data. **b** Performance comparison between scCASER and scCASE. **c** The clustering performance of scCASER on the Spleen dataset with different numbers of self-reference pseudo-bulk samples. The dashed line represents the performance of scCASE method. Source data are provided as a Source Data file.

an input, then utilizes NMF to reduce the dimension of reference data and extract prior knowledge as a part of the projection matrix ("Methods").

Integrating reference data in scCASER can better characterize the target scCAS data, enrich the enhancement process, and improve the quality of downstream analyses. To expand the applicability of scCA-SER, we explored various methods to obtain reference data ("Methods"). First, for the Blood and BM0828 datasets, similar to existing studies[7,21], we obtained the reference from the bulk data encompassing 17 distinct blood cell types[23]. Secondly, for the paired datasets of the same tissue, i.e., WholeBrainA/WholeBrainB and LungA/LungB, we generated pseudo-bulk reference for one of the datasets via aggregating the count matrix of another dataset by cell type labels. Finally, in cases where datasets have no related bulk data or paired single-cell dataset available, such as the LargeIntestine and Spleen datasets, we used a general scCAS analysis pipeline to cluster cells[26], aggregated the target scCAS count matrix by clustering labels, and obtained pseudo-bulk self-reference data. With the strategy, we could leverage the inherent information of target scCAS data to create representative reference data without introducing external information. The various strategies for obtaining reference data enable us to expand the applicability of scCASER.

We assessed scCASER's performance in improving cell clustering and estimation of cellular distances. The results indicated that upon integrating reference data, the values of different metrics overall exhibited obvious improvement compared with scCASE without incorporating reference data. The average improvement was 9.05% in ARI, 4.04% in AMI, 5.70% in FMI, and 3.19% in silhouette score (Fig. 6b;

Supplementary Figs. S2–S4 and Supplementary Table S2). Indeed, different reference data may impact the accuracy of the enhancement of scCASER. Taking the Spleen dataset as an example, the reference data generated by Louvain clustering with default parameters initially comprised 11 clusters, providing comprehensive reference data. If we only consider a subset of these clusters, the resulting reference data will contain less information. To illustrate the impact of different reference data for scCASER, we dropped out the clusters in sequence from 11 clusters and obtained reference data containing 8, 6, and 4 clusters, respectively. In this scenario, we observed that scCASER can still benefit from the information provided by the incomplete reference data, while their performances are comparatively inferior to that achieved with the complete reference data containing all 11 clusters (Fig. 6c). Despite this, scCASER still surpasses the performance of scCASE (the orange dashed line). The results indicate that while reference data with incomplete information may lead to a slight decline in performance compared with the complete one, incorporating reference data in scCASER can generally prove advantageous for scCAS data enhancement.

## Discussion

In this study, we propose a non-negative matrix factorization-based scCAS data enhancement method called scCASE. Through comprehensive experiments on multiple datasets generated with different protocols, from various species and tissues, and of divergent sizes, dimensions, and qualities, we have demonstrated the superior performance of scCASE over state-of-the-art methods in imputing dropout events in scCAS data and thus facilitating downstream analyses.

Moreover, we have shown that the enhancement process of scCASE is interpretable, making scCASE a valuable tool for elucidating cellular heterogeneity and revealing cell type-specific biological insights. In addition, we developed scCASER, which enables the incorporation of external data as a reference to better characterize the target scCAS data and facilitate data enhancement. We also introduced multiple strategies to obtain or construct reference data, thereby expanding the applicability of our model in various scenarios.

Certainly, while scCASE offers significant advancements, it does exhibit certain limitations. We provide several directions for further improving scCASE. First, the current model relies upon linear products of non-negative matrices, which presents challenges in capturing nonlinear patterns. We can introduce nonlinear projection through sophisticated techniques like deep non-negative matrix factorization or integration with neural networks. Second, although we have proposed a strategy to choose the optimum step size in gradient descent, scCASE only has comparable performance in computational efficiency compared with other approaches. This drawback can be mitigated by employing GPU for parallel operations, thereby accelerating the execution of scCASE. Third, the application of the modeling ideas in scCASE may not be confined solely to scCAS data. Given the accuracy and interpretability of scCASE, we speculate the prosperous applications to single-cell data of other omics, such as spatial omics and even multi-omics.

## Methods

### The model of scCASE

Given a scCAS count matrix, we first filter out the peaks expressed in fewer than 1% of the cells (Supplementary Text S6)[7,9,59], and then use TF-IDF transformation to reweight peaks by their occurrence frequencies[7,59]. The formula of TF-IDF transformation is shown in Eq. (1):

$$x_{ij} = \left(\frac{\hat{x}_{ij}}{\sum_{k=1}^{m} \hat{x}_{kj}}\right) \cdot \log\left(\frac{n}{\sum_{t=1}^{n} \hat{x}_{it}}\right) \tag{1}$$

where $\hat{x}_{ij}$ is the read count in peak $i$ of cell $j$. We denote the preprocessed count matrix as $\mathbf{X} \in \mathbb{R}^{m \times n}$, where $m$ is the number of peaks and $n$ is the number of cells. We then minimize the loss function $F$ in Eq. (2) to enhance the data in an iterative manner:

$$\min_{\mathbf{W},\mathbf{H},\mathbf{Z} \geq 0} F = \|\mathbf{X}(\mathbf{Z} \circ \mathbf{R}) - \mathbf{WH}\|_F^2 + \lambda\|\mathbf{Z} - \mathbf{H}^T\mathbf{H}\|_F^2 + \gamma_1\|\mathbf{W}\|_F^2 + \gamma_2\|\mathbf{H}\|_F^2 \tag{2}$$

where $\mathbf{Z} \in \mathbb{R}^{n \times n}$ is the cell-to-cell similarity matrix in which each column sums to 1, and $z_{ij}$ represents the similarity between cell $i$ and cell $j$. $\mathbf{R} \in \mathbb{R}^{n \times n}$ is the random sampling matrix, where each element is either 0 or 1 generated through binomial distribution. $\mathbf{R}$ is Hadamard multiplied with $\mathbf{Z}$ in the computation to avoid the similar cells exhibiting almost the same accessible peaks which improperly reduces the cellular heterogeneity. $\mathbf{W} \in \mathbb{R}^{m \times k}$ is the factorized projection matrix while $\mathbf{H} \in \mathbb{R}^{k \times n}$ is the corresponding cell embedding matrix.

More specifically, the scCASE model can be divided into two parts. On the one hand, $\mathbf{X}(\mathbf{Z} \circ \mathbf{R})$ is the matrix enhanced by similarity. On the other hand, $\mathbf{WH}$ is the matrix reconstructed by matrix factorization. We aim to minimize the difference between the two matrices. The first term in Eq. (2) uses the Frobenius norm to approach $\mathbf{X}(\mathbf{Z} \circ \mathbf{R})$ and $\mathbf{WH}$. The second term minimizes the Frobenius norm of the cell-to-cell similarity matrix and the product of $\mathbf{H}^T$ and $\mathbf{H}$. This term makes $\mathbf{Z}_{ij}$ similar to the Cartesian product of the cell embedding vectors of cell $i$ and cell $j$. Similar cells should have smaller angles between their embedding vectors and more oversized Cartesian products, corresponding to larger elements in the similarity matrix. The third and fourth terms represent two regularization terms that constrain the projection matrix $\mathbf{W}$ and the cell embedding $\mathbf{H}$, respectively, to

prevent model overfitting. The Frobenius norm is commonly used in non-negative matrix factorization due to its differentiability, which facilitates the optimization process. Coefficients of $\lambda$, $\gamma_1$, and $\gamma_2$ serve as weights of different terms. We also discuss the differences and advantages of scCASE to PCA/SVD in Supplementary Text S7.

### The model of scCASE with reference data

We also introduce a variant of scCASE, named scCASER, to incorporate available omics data as reference data to further improve the data enhancement performance. The data preprocessing strategy in scCASER is the same as that in scCASE. We perform conventional NMF on the reference data and obtain the factorized projection matrix $\mathbf{P} \in \mathbb{R}^{m \times k_1}$. We introduce the projection matrix learned from reference data to the loss function $F$, as shown in Eq. (3):

$$\min_{\mathbf{W}_{1,2},\mathbf{H},\mathbf{Z} \geq 0} F = \|\mathbf{X}(\mathbf{Z} \circ \mathbf{R}) - [\mathbf{W}_1,\mathbf{W}_2]\mathbf{H}\|_F^2 + \lambda\|\mathbf{Z}$$
$$- \mathbf{H}^T\mathbf{H}\|_F^2 + \gamma_1\|\mathbf{W_m}\|_F^2 + \gamma_2\|\mathbf{H}\|_F^2 + \alpha\|\mathbf{P} - \mathbf{W}_1\|_F^2 \tag{3}$$

The projection matrix $\mathbf{W_m} \in \mathbb{R}^{m \times k}$ is decomposed into two parts and can be written as $[\mathbf{W_1},\mathbf{W_2}]$, where $\mathbf{W}_1 \in \mathbb{R}^{m \times k_1}$ is used to transfer the epigenetic information from reference data, while $\mathbf{W}_2 \in \mathbb{R}^{m \times k_2}$ is the projection matrix learned during fitting the target scCAS data. Additionally, $k_1$ and $k_2$ are the numbers of columns in $\mathbf{W}_1$ and $\mathbf{W}_2$, respectively ($k = k_1 + k_2$), and they can be adjusted to control the dominance of the reference data on the model. The last regularization term of the optimization problem imposes a constraint on $\mathbf{W}_1$ using the matrix $\mathbf{P}$, to mitigate the difference between the projection matrix learned from reference data and the projection matrix learned from target data. Other settings of scCASER are similar to that of scCASE. Coefficients of $\lambda$, $\gamma_1$, $\gamma_2$, and $\alpha$ serve as weights of different terms.

### Iterative optimization of scCASE

For scCASE, to minimize the loss function in Eq. (2), we can update the three matrices $\mathbf{W}$, $\mathbf{H}$, and $\mathbf{Z}$ iteratively using the gradient descent algorithm. To achieve this, we first compute the partial derivatives of the loss function with respect to each matrix. According to the definition of the Frobenius norm and the trace of a matrix, we can convert the loss function from a norm form to a trace form as Eq. (4), making it easier to compute the gradient:

$$F = tr((\mathbf{Z} \circ \mathbf{R})^T\mathbf{X}^T\mathbf{X}(\mathbf{Z} \circ \mathbf{R}) - \mathbf{H}^T\mathbf{W}^T\mathbf{X}(\mathbf{Z} \circ \mathbf{R}) - (\mathbf{Z} \circ \mathbf{R})^T\mathbf{X}^T\mathbf{WH} + \mathbf{H}^T\mathbf{W}^T\mathbf{WH})$$
$$+ \lambda tr(\mathbf{Z}^T\mathbf{Z} - \mathbf{H}^T\mathbf{HZ} - \mathbf{Z}^T\mathbf{H}^T\mathbf{H} + \mathbf{H}^T\mathbf{HH}^T\mathbf{H}) + \gamma_1 tr(\mathbf{W}^T\mathbf{W}) + \gamma_2 tr(\mathbf{H}^T\mathbf{H}) \tag{4}$$

After obtaining the partial derivatives of $F$ with respect to $\mathbf{W}$, $\mathbf{H}$, and $\mathbf{Z}$ (Eq. (5)), we use gradient descent to optimize the model:

$$\frac{\partial F}{\partial \mathbf{W}} = -2X(Z \circ R)\mathbf{H}^T + 2WH\mathbf{H}^T + 2\gamma_1 W$$

$$\frac{\partial F}{\partial \mathbf{H}} = -2\mathbf{W}^TX(Z \circ R) + 2\mathbf{W}^TWH - 2\lambda H(\mathbf{Z} + \mathbf{Z}^T) + 4\lambda H\mathbf{H}^TH + 2\gamma_2 H$$

$$\frac{\partial F}{\partial \mathbf{Z}} = 2(\mathbf{X}^TX(\mathbf{Z} \circ \mathbf{R})) \circ R - 2(\mathbf{X}^TWH) \circ R + 2\lambda Z - 2\lambda\mathbf{H}^TH \tag{5}$$

The iteration will stop if the change between two consecutive iterations is less than $10^{-6}$. Finally, we multiply the raw data matrix $\mathbf{X}$ by the iteratively optimized cell-cell similarity matrix $\mathbf{Z}$, resulting the enhanced scCAS data $\mathbf{XZ}$.

To improve the efficiency of the optimization algorithm, we propose a strategy to choose the optimal step size for each iteration. We let $\frac{\partial F}{\partial \mathbf{W}} = \mathbf{D}_1$, $\frac{\partial F}{\partial \mathbf{Z}} = \mathbf{D}_2$ and $\frac{\partial F}{\partial \mathbf{H}} = \mathbf{D}_3$. For $\mathbf{W}$, we assume the most appropriate step size $\delta_1$ should be the one that minimizes

$F(\mathbf{W} - \delta_1\mathbf{D_1}, \mathbf{H}, \mathbf{Z})$, as shown in Eq. (6):

$$F = tr((\mathbf{Z} \circ \mathbf{R})^T \mathbf{X}^T \mathbf{X}(\mathbf{Z} \circ \mathbf{R}) - \mathbf{H}^T(\mathbf{W} - \delta_1\mathbf{D_1})^T \mathbf{X}(\mathbf{Z} \circ \mathbf{R}))$$
$$+ tr(-(\mathbf{Z} \circ \mathbf{R})^T \mathbf{X}^T(\mathbf{W} - \delta_1\mathbf{D_1})\mathbf{H} + \mathbf{H}^T(\mathbf{W} - \delta_1\mathbf{D_1})^T(\mathbf{W} - \delta_1\mathbf{D_1})\mathbf{H})$$
$$+ \lambda tr(\mathbf{Z}^T \mathbf{Z} - \mathbf{H}^T \mathbf{H} \mathbf{Z} - \mathbf{Z}^T \mathbf{H}^T \mathbf{H} + \mathbf{H}^T \mathbf{H} \mathbf{H}^T \mathbf{H}) + \gamma_1 tr(\mathbf{W}^T \mathbf{W}) + \gamma_2 tr(\mathbf{H}^T \mathbf{H})$$

(6)

By substituting the parameters into Eq. (6) and taking the derivative, we can obtain the following:

$$\frac{dF}{d\delta_1} = tr(\mathbf{H}^T \mathbf{D_1}^T \mathbf{X}(\mathbf{Z} \circ \mathbf{R})) + tr((\mathbf{Z} \circ \mathbf{R})^T \mathbf{X}^T \mathbf{D_1}\mathbf{H}) - tr(\mathbf{H}^T \mathbf{W}^T \mathbf{D_1}\mathbf{H})$$
$$- tr(\mathbf{H}^T \mathbf{D_1}^T \mathbf{W}\mathbf{H}) + 2\delta_1 tr(\mathbf{H}^T \mathbf{D_1}^T \mathbf{D}\mathbf{H})$$

(7)

When obtaining the minimum value of $F$, the derivative of $F$ with respect to $\delta$ is zero. Let Eq. (7) equal to 0, and we can solve the value $\delta_1$ as:

$$\delta_1 = \frac{-tr(\mathbf{H}^T \mathbf{D_1}^T \mathbf{X}(\mathbf{Z} \circ \mathbf{R})) - tr((\mathbf{Z} \circ \mathbf{R})^T \mathbf{X}^T \mathbf{D_1}\mathbf{H}) + tr(\mathbf{H}^T \mathbf{W}^T \mathbf{D_1}\mathbf{H}) + tr(\mathbf{H}^T \mathbf{D_1}^T \mathbf{W}\mathbf{H})}{2tr(\mathbf{H}^T \mathbf{D_1}^T \mathbf{D_1}\mathbf{H})}$$

(8)

For $\mathbf{Z}$, we suppose the best step size is $\delta_2$, and then the $\delta_2$ will let $F(\mathbf{W}, \mathbf{H}, \mathbf{Z} - \delta_2\mathbf{D_2})$ get minimum, $\frac{dF(\mathbf{W}, \mathbf{H}, \mathbf{Z} - \delta_2\mathbf{D_2})}{d\delta_2} = 0$, and the calculation of loss function $F$ is as below:

$$F = tr(((\mathbf{Z} - \delta_2\mathbf{D_2}) \circ \mathbf{R})^T \mathbf{X}^T \mathbf{X}(\mathbf{Z} - \delta_2\mathbf{D_2}) \circ \mathbf{R} - \mathbf{H}^T \mathbf{W}^T \mathbf{X}(\mathbf{Z} - \delta_2\mathbf{D_2}) \circ \mathbf{R})$$
$$+ tr(-((\mathbf{Z} - \delta_2\mathbf{D_2}) \circ \mathbf{R})^T \mathbf{X}^T \mathbf{W}\mathbf{H} + \mathbf{H}^T \mathbf{W}^T \mathbf{W}\mathbf{H})$$
$$+ \lambda tr((\mathbf{Z} - \delta_2\mathbf{D_2})^T(\mathbf{Z} - \delta_2\mathbf{D_2}) - \mathbf{H}^T \mathbf{H}(\mathbf{Z} - \delta_2\mathbf{D_2}) - (\mathbf{Z} - \delta_2\mathbf{D_2})^T \mathbf{H}^T \mathbf{H}$$
$$+ \mathbf{H}^T \mathbf{H} \mathbf{H}^T \mathbf{H}) + \gamma_1 tr(\mathbf{W}^T \mathbf{W}) + \gamma_2 tr(\mathbf{H}^T \mathbf{H})$$

(9)

$$\frac{dF}{d\delta_2} = 0 = -tr((\mathbf{D_2} \circ \mathbf{R})^T \mathbf{X}^T \mathbf{X}(\mathbf{Z} \circ \mathbf{R}) + (\mathbf{Z} \circ \mathbf{R})^T \mathbf{X}^T \mathbf{X}(\mathbf{D_2}\mathbf{R}))$$
$$+ 2\delta_2 tr((\mathbf{D_2} \circ \mathbf{R})^T \mathbf{X}^T \mathbf{X}(\mathbf{D_2} \circ \mathbf{R}))$$
$$+ tr(\mathbf{H}^T \mathbf{W}^T \mathbf{X}(\mathbf{D_2} \circ \mathbf{R}) + (\mathbf{D_2} \circ \mathbf{R})^T \mathbf{X}^T \mathbf{W}\mathbf{H})$$
$$- \lambda tr(\mathbf{Z}^T \mathbf{D_2} + \mathbf{D_2}^T \mathbf{Z}) + 2\lambda\delta_2 tr(\mathbf{D_2}^T \mathbf{D_2})$$
$$+ \lambda tr(\mathbf{H}^T \mathbf{H}\mathbf{D_2} + \mathbf{D_2}^T \mathbf{H}^T \mathbf{H})$$

(10)

Then, we get the optimal $\delta_2$ value as Eq. (11):

$$\delta_2 = \frac{tr(\mathbf{eq1}) - tr(\mathbf{eq3}) + \lambda tr(\mathbf{eq4}) - \lambda tr(\mathbf{eq6})}{2tr(\mathbf{eq2}) + 2\lambda tr(\mathbf{eq5})}$$

(11)

where

$$\mathbf{eq1} = (\mathbf{D_2} \circ \mathbf{R})^T \mathbf{X}^T \mathbf{X}(\mathbf{Z} \circ \mathbf{R}) + (\mathbf{Z} \circ \mathbf{R})^T \mathbf{X}^T \mathbf{X}(\mathbf{D_2} \circ \mathbf{R})$$
$$\mathbf{eq2} = (\mathbf{D_2} \circ \mathbf{R})^T \mathbf{X}^T \mathbf{X}(\mathbf{D_2} \circ \mathbf{R})$$
$$\mathbf{eq3} = \mathbf{H}^T \mathbf{W}^T \mathbf{X}(\mathbf{D_2} \circ \mathbf{R}) + (\mathbf{D_2} \circ \mathbf{R})^T \mathbf{X}^T \mathbf{W}\mathbf{H}$$
$$\mathbf{eq4} = \mathbf{Z}^T \mathbf{D_2} + \mathbf{D_2}^T \mathbf{Z}$$
$$\mathbf{eq5} = \mathbf{D_2}^T \mathbf{D_2}$$
$$\mathbf{eq6} = \mathbf{H}^T \mathbf{H}\mathbf{D_2} + \mathbf{D_2}^T \mathbf{H}^T \mathbf{H}$$

(12)

At each iteration, the optimal step size will be calculated and then used to update the variable matrix, accelerating convergence speed and reducing the computational cost. For scCASE, the optimal step size for $\mathbf{W}$ and $\mathbf{Z}$ can be adaptively obtained, while for $\mathbf{H}$, due to its high order terms, we cannot find the optimal step size in a similar way. We

initialize the step size $\delta_3$ of $\mathbf{H}$ to 0.2. If this step size cannot reduce the loss function during the iteration process, we will reduce the step size and try updating $\mathbf{H}$ again.

In a similar fashion to the scCASE approach, we additionally introduce two mask matrices of $\mathbf{M}$ and $\mathbf{N}$ when implementing scCASER, and Eq. (3) becomes:

$$\min_{\mathbf{W}_{1,2}, \mathbf{H}, \mathbf{Z} \geq 0} F = \|\mathbf{X}(\mathbf{Z} \circ \mathbf{R}) - (\mathbf{M} \circ \mathbf{W}_1 + \mathbf{N} \circ \mathbf{W}_2)\mathbf{H}\|_F^2 + \lambda\|\mathbf{Z} - \mathbf{H}^T \mathbf{H}\|_F^2$$
$$+ \gamma_1\|\mathbf{W_m}\|_F^2 + \gamma_2\|\mathbf{H}\|_F^2 + \alpha\|\mathbf{P} - \mathbf{W}_1\|_F^2$$

(13)

In Eq. (13), $\mathbf{W}_1$ is expanded from the original $\mathbf{W}_1$ to $[\mathbf{W}_1, \mathbf{0}]$ to have the same dimension as $\mathbf{W_m}$. $\mathbf{M}$ is a masking matrix with 1 in the region where $\mathbf{W}_1$ acts and 0 elsewhere. The same is done for $\mathbf{W}_2$ and $\mathbf{N}$. Hadamard multiplying $\mathbf{M}$, $\mathbf{N}$ by $\mathbf{W}_1$, $\mathbf{W}_2$ respectively, and adding them yields the equivalent effect of Eq. (3). The loss function $F$ can then be written as Eq. (14):

$$F = tr((\mathbf{Z} \circ \mathbf{R})^T \mathbf{X}^T \mathbf{X}(\mathbf{Z} \circ \mathbf{R}) - \mathbf{H}^T(\mathbf{M} \circ \mathbf{W}_1 + \mathbf{N} \circ \mathbf{W}_2)^T \mathbf{X}(\mathbf{Z} \circ \mathbf{R})$$
$$- (\mathbf{Z} \circ \mathbf{R})^T \mathbf{X}^T(\mathbf{M} \circ \mathbf{W}_1 + \mathbf{N} \circ \mathbf{W}_2)\mathbf{H}$$
$$+ \mathbf{H}^T(\mathbf{M} \circ \mathbf{W}_1 + \mathbf{N} \circ \mathbf{W}_2)^T(\mathbf{M} \circ \mathbf{W}_1 + \mathbf{N} \circ \mathbf{W}_2)\mathbf{H}))$$
$$+ \lambda tr(\mathbf{Z}^T \mathbf{Z} - \mathbf{H}^T \mathbf{H}\mathbf{Z} - \mathbf{Z}^T \mathbf{H}^T \mathbf{H} + \mathbf{H}^T \mathbf{H}\mathbf{H}^T \mathbf{H})$$
$$+ \gamma_1 tr(\mathbf{W_m}^T \mathbf{W_m}) + \gamma_2 tr(\mathbf{H}^T \mathbf{H})$$
$$+ \alpha tr(\mathbf{P}^T \mathbf{P} - (\mathbf{M} \circ \mathbf{W}_1)^T \mathbf{P} - \mathbf{P}^T(\mathbf{M} \circ \mathbf{W}_1) + (\mathbf{M} \circ \mathbf{W}_1)^T(\mathbf{M} \circ \mathbf{W}_1))$$

(14)

Details of the optimization process for scCASER are similar to that of scCASE and can be found in Supplementary Texts S2.

### Initialization and parameter selection of scCASE

Due to the high-dimensional characteristic of the variable matrix, random initialization cannot guarantee that the algorithm converges quickly to the desired solution. Therefore, we initialize $\mathbf{W}$, $\mathbf{H}$, and $\mathbf{Z}$ specifically. We initialize $\mathbf{Z}$ as the Jaccard similarity matrix between cells[18], and $\mathbf{W}$ and $\mathbf{H}$ as the projection matrix and cell embedding matrix obtained by performing conventional NMF on the target scCAS matrix $\mathbf{X}$, respectively. In scCASER, $\mathbf{W}_1$ is initialized as the projection matrix obtained by NMF on the reference data and $\mathbf{W}_2$ is initialized as the projection matrix obtained by NMF on the target scCAS data. We use ASTER to estimate the number of cell types as the number of latent factors in non-negative matrix factorization, which is empirically suitable for various scCAS datasets[24]. The default value of lambda in the model is $10^6$.

As pointed out in the existing literature, enhancement may lead to over-smoothing, resulting in the removal of true cell-cell heterogeneity signals[11,60]. We validated the impact of different initializations and parameter choices on over-smoothing. We designed three additional metrics, namely, over-smoothing score, under-smoothing score, and smoothing score (Supplementary Text S1), to assess the degree of over-smoothing. First, for model initialization, random initialization of $\mathbf{H}$ would severely affect the matrix $\mathbf{Z}$ and lead to unsatisfactory outcomes (Supplementary Fig. S24). The model with random initialization of $\mathbf{Z}$ will still convergent, though leading to a worse performance of enhancement, and the random initialization of $\mathbf{Z}$ does not result in over-smoothing (Supplementary Fig. S24). Secondly, we ran scCASE with varying lambda values within the range of $10^5$ to $10^8$. As lambda varies, different metrics remain stable with few changes (Supplementary Fig. S25). This indicates the high robustness of scCASE to the choice of lambda, suggesting that within a certain range, the choice of lambda does not lead to over-smoothing. Finally, we validated the impact of different values of parameter $K$. The results indicate that when $K$ is small ($K<7$), it can be observed that at lower dimensions, the model struggles to capture differences in the data effectively, leading to the elimination

of heterogeneity between different cell types and over-smoothing of the data (Supplementary Figs. S26–S28). As $K$ gradually increases, the degree of over-smoothing eases. However, though not lead to over-smoothing, large values of $K$ ($K>20$) may introduce excessive noise, making the model learning more challenging and resulting in a lower under-smoothing score (Supplementary Figs. S26–S28).

## Run-time and memory usage of scCASE

scCASE consistently exhibited commendable performance of run-time and peak memory usage. In terms of run-time, scCASE exhibits significant advantages compared to other methods, especially on smaller datasets such as BM0828, Blood, and LungA, where scCASE can operate several times faster than baseline methods (Supplementary Fig. S29). Even on larger datasets, scCASE still maintains a notable speed advantage. In terms of peak memory usage, SCALE and scBasset are GPU-based methods, they make more usage of GPU memory, so their memory usage is typically smaller than the methods that utilize CPUs. scCASE demonstrates a certain advantage in peak memory usage on smaller datasets such as BM0828, Blood, and LungA (Supplementary Fig. S29). Although the peak memory usage of scCASE increases in larger datasets, its memory usage is still comparable to that of scOpen, the state-of-the-art scCAS data enhancement method (Supplementary Fig. S29). Moreover, the memory usage of scCASE growth remains manageable. On two larger datasets of Muto and Simulated, which have a similar number of peaks but a fivefold increase in the number of cells (from 20k to 100k), the peak memory usage of scCASE increased by 7.67 times, while that of scOpen increased by 8.62 times. Note that scBFA is unable to run on datasets with 100k cells due to out-of-memory errors.

## Implementation details of baseline methods

scBFA: scBFA is a detection-based model to remove technical variation in scRNA-seq and scATAC-seq data, available at https://github.com/quon-titative-biology/scBFA[16]. We utilized the raw count matrices as input and performed scBFA using their default parameters. We executed scBFA following the same benchmarking procedure as in scOpen (https://github.com/CostaLab/scopen-reproducibility/blob/main/scripts/Imputation/scBFA.R).

SCALE: SCALE integrates both the variational auto-encoder (VAE) and the Gaussian mixture model (GMM) to characterize the distribution of scATAC-seq data[9]. We obtained the SCALE program from https://github.com/jsxlei/SCALE and used the raw count matrices as input. When executing the program, we set the option "impute" as TRUE to obtain the imputed data, while keeping other parameters at their default settings.

scBasset: scBasset is a sequence-based convolutional neural network method to model scATAC data and predict chromatin accessibility[17]. The program and training tutorial of scBasset can be downloaded from https://github.com/calico/scBasset. We trained scBasset with default parameters using the raw count matrices and peaks as the input. Genome fasta file used in scBasset can be downloaded from (https://hgdownload.soe.ucsc.edu/downloads.html). After obtaining the trained model, we referred to their tutorial to obtain the enhanced data (https://github.com/calico/scBasset/blob/main/examples/PBMC_multiome/evaluate.ipynb).

scOpen: scOpen is a scCAS-seq imputation method based on regularized non-negative matrix factorization[10]. The raw scCAS count matrix serves as the input for scOpen, and the output is an imputed matrix. We followed the tutorial and examples of scOpen provided at https://github.com/CostaLab/scopen and executed it with default parameters.

## Implementation details of downstream analyses

t-SNE and UMAP Visualization: we first preprocessed the raw data and the data enhanced by different methods using TF-IDF. Then, we performed PCA to reduce the dimensionality, created a "neighbors" graph, and used t-SNE/UMAP to obtain two-dimensional visualizations of the data, respectively following the scATAC-seq data analysis workflow provided by EpiScanpy[26] (https://colomemaria.github.io/episcanpy_doc/examples.html). The above steps were performed using the default parameters in the EpiScanpy pipeline.

SNPsea: SNPsea is an algorithm to identify cell types and pathways likely to be affected by risk loci. Specifically, genome-wide association studies (GWAS) have discovered multiple genomic loci associated with risk for different types of disease. SNPsea provides a simple way to determine the types of cells influenced by genes in these risk loci. SNPsea supposes disease-associated alleles influence a small number of pathogenic cell types, and assumes that a gene's specificity to a cell type is a reasonable indicator of its importance to the unique function of that cell type. We performed SNPsea analysis with default settings in each set of cell type-specific peaks and the set of background peaks, respectively. The enrichments of tissue-specific expression in profiles of 17,581 genes across 79 human tissues (Gene Atlas) were quantified[61]. Specifically, we first obtained SNP site data for the whole genome from HapMap3 SNPs, which can be downloaded at https://zenodo.org/records/7768714. To obtain the SNP sites corresponding to each group of cell type-specific peaks, we utilized GenomicRanges to identify SNP sites present within the cell type-specific peak regions[62]. GenomicRanges is an R/Bioconductor package for representing and manipulating genomic intervals, available at https://github.com/Bioconductor/GenomicRanges. Then we obtained the SNP sites corresponding to each group of cell type-specific peaks, which serve as the input for SNPsea. We specified the same additional data including phenotype data and parameters for SNPsea as in its tutorial. These additional data can be downloaded from http://www.broadinstitute.org/mpg/snpsea (SNPsea_data_20140520.zip). Their sources and detailed explanations are described at https://snpsea.readthedocs.io/en/latest/data.html. The parameters and specific running tutorials of the method can be found at https://snpsea.readthedocs.io/en/latest/usage.html. We quantified the enrichments of each set of peaks in tissue-specific accessibility profiles across 79 tissues, and the top 30 significantly enriched tissues are illustrated in the figures.

LDSC: LDSC is a command line tool for estimating heritability and genetic correlation from GWAS summary statistics[40]. After identifying cell type-specific peaks and background peaks in the Blood dataset, we quantified the enrichment of heritability for blood-related phenotypes within cell type-specific peaks for each cell type using partitioned LDSC with default settings. We ran LDSC using HapMap3 SNPs and used European samples from the 1000 Genomes Project as the LD reference panel. All the summary statistics provided by LDCS including SNPs and phenotypes were downloaded from the Broad LD Hub (https://doi.org/10.5281/zenodo.7768714). Specifically, in this analysis, our input consists of the detected cell type-specific peaks or background peaks. Similar to SNPsea, we first utilized GenomicRanges to identify SNP sites present within the specific peak regions. Subsequently, we used the LDSC program to calculate the LD Scores for these SNP sites. The LDSC process in this step can be referred to at https://github.com/bulik/ldsc/wiki/LD-Score-Estimation-Tutorial. Finally, we invoked the LDSC program again, using the obtained LD Scores as input, to calculate heritability and genetic correlation with blood-related phenotypes. The LDSC process in this step can be referred to https://github.com/bulik/ldsc/wiki/Heritability-and-Genetic-Correlation.

scABC: scABC is an R package for the analysis of scATAC-seq data[41]. With the clustering assignments obtained from scCASE and Louvain clustering, we followed the scABC workflow, utilized the function "getClusterSpecificPvalue()", calculated the $p$ value using hypothesis testing procedure, and finally identified cluster-specific peaks (https://github.com/SUwonglab/scABC/blob/master/vignettes/ExampleWorkflow.html). These identified cluster-specific peaks will be

used as the input of chromVAR[42] to perform motif analysis similar to RA3[7].

chromVAR: chromVAR is an R package for the analysis of sparse chromatin accessibility data from single-cell/bulk ATAC-seq/DNase-seq[42]. The package aims to identify motifs or other genomic annotations associated with variability in chromatin accessibility between individual cells or samples. We downloaded chromVAR from https://greenleaflab.github.io/chromVAR/. The motifs database is obtained from the "getJasparMotifs()" function within the chromVAR[42] method, which is sourced from the JASPAR[63] database. Following the workflow in RA3[7], we used the data enhanced by scCASE as the input and applied chromVAR[42] to infer the enriched transcription factor (TF) binding motifs within the top 1000 cluster-specific peaks with the smallest $p$ values calculated by scABC. Subsequently, we visualized the deviations calculated by chromVAR for the top 50 TF binding motifs.

## Data collection and preprocessing

We utilized 13 datasets in this study, including a simulated scCAS dataset, eight publicly available scCAS datasets, two scCAS datasets annotated based on paired scRNA-seq data, and two mixed scCAS datasets (Supplementary Table S1). The simulated scCAS dataset is created by simCAS[22], the state-of-the-art method for scCAS data simulation, in discrete mode to construct a dataset consisting of five clusters, each containing 500 cells. We then used the EpiScanpy pipeline to identify the top 3000 differentially accessible peaks for each cell type[26]. In this way, we obtained a simulated dataset with 2500 cells and 15,000 peaks. We collected eight real scCAS datasets for benchmarking. The Blood dataset contains ten types of human hemocytes from bone marrow, while the BM0828 dataset is a subset of the Blood dataset with the donor label of BM0828, containing seven types of human hemocytes[23]. To investigate the applicability of methods to datasets from different species and tissues, we further collected six datasets of WholeBrainA, WholeBrainB, LungA, LungB, LargeIntestine, and Spleen, which were profiled from the whole brain, lung, large intestine, and spleen tissues of adult mice[25]. These datasets vary in terms of protocol, species, tissue, number of cells, number of cell types, number of peaks, and imbalance degree, but all exhibit high sparsity and dimension (Supplementary Table S1). For the two scCAS datasets which were annotated solely based on paired scRNA-seq data, the Muto dataset was profiled via snATAC-seq and snRNA-seq and contains human kidney cells[64]. The snRNA-seq dataset was annotated based on lineage-specific marker expression, and the annotated snRNA-seq dataset was leveraged to predict snATAC-seq cell types with the label transfer function in Seurat[65]. The PBMC dataset was profiled by "10x Genomics via Single Cell Multiome ATAC + Gene Expression Sequencing" and contains the cryopreserved human peripheral blood mononuclear cells (PBMCs) of a healthy female donor. The cell type labels were annotated in the original study using only the scRNA-seq data. The Mixed-tissues and Mixed-protocols datasets are obtained by concatenating datasets from various tissues and protocols, respectively, to evaluate the robustness of different methods. The Mixed-tissues dataset was generated from the LungA and Spleen datasets while the Mixed-protocols dataset contains two batches assayed by snATAC-seq and 10X[55,56]. Cell types that accounted for less than three percent of the total cells were discarded along with the unknown categories to ensure evaluation credibility. We determine the imbalance degree by estimating the normalized entropy of the cell-type size distribution[24] using Eq. (15):

$$I = 1 + \frac{1}{\log C} \sum_{c=1}^{C} \frac{n_c}{N} \log \frac{n_c}{N} \qquad (15)$$

where $C$ is the number of cell types in the dataset, $n_c$ is the number of cells in the cell type $c$ and $N$ is the total number of cells in the dataset. The more imbalanced the dataset is, the higher the value $I$ has. This metric will have a value of 1 if all the cells have the same type and 0 if all the cell types have the same number of cells.

## Construction of reference data

The construction approaches of reference data for scCASER are flexible and diverse. Firstly, we can obtain references from existing bulk data. For the Blood and BM0828 datasets, the reference data was constructed using bulk data from 17 hematopoietic cell types[23]. We counted the reads aligned to peaks of the scCAS data for the bulk samples, resulting in a count matrix that shares the same peaks as the target scCAS data. Then, we obtained the reference data by scaling the count matrix based on the total mapped reads of each bulk sample[7].

Secondly, we can construct pseudo-bulk reference data by aggregating scCAS data of an external dataset from a similar tissue as the target data. The WholeBrainA and WholeBrainB datasets, as well as the LungA and LungB datasets, are paired data derived from similar tissues. We grouped the cells in one of the paired datasets by cell type, took the sum of each peak over cells of the same type, and then obtained the pseudo-bulk reference data for another dataset.

Thirdly, we can aggregate the scCAS data by its own clustering labels to construct the pseudo-bulk self-reference data. For scCAS datasets without external reference data, such as the Spleen and LargeIntestine datasets, we applied Louvain clustering with the default resolution, took the sum of each peak over cells of the same cluster, and then obtained the pseudo-bulk self-reference data for the target scCAS data.

## Performance evaluation

We evaluated data enhancement performance via numerical accuracy, cell clustering, and data visualization. For numerical accuracy, we utilized the area under the precision-recall curve (auPRC) and the area under the receiver operating characteristic curve (auROC) to test if the enhanced matrix can recover the true signal cell-wise and peak-wise, respectively[66]. auPRC considers the relationship between precision and recall, while auROC considers the relationship between correctly classified positive examples and the number of incorrectly classified negative examples. auPRC is preferred for tasks with a large skew in the class distribution.

For cell clustering, we adopted the widely-used Louvain algorithm[26,65,67,68] and utilized a binary search strategy to ensure the number of clusters equals the number of cell types[7,24,26,27,69]. We assessed the clustering results by four widely used metrics: adjusted Rand index (ARI)[28], adjusted mutual information (AMI)[29], Fowlkes-Mallows index(FMI)[30], and silhouette score[31]. Rand index (RI) computes a similarity measure between the cluster labels and the cell-type labels. ARI is adjusted based on RI and accounts for chance agreement. Mutual information (MI) quantifies the correlation between the cluster labels and the cell-type labels and NMI is a normalized variant of MI. The Fowlkes-Mallows score FMI is defined as the geometric mean of the pairwise precision and recall. The silhouette score measures the similarity between an object and its own cluster compared to that of other clusters. While silhouette score is commonly utilized clustering labels, we replaced it with the cell type labels to evaluate the performance of the impact of enhancement on the estimation of the distance between cells as scOpen[10], and the higher the silhouette score, the better the performance. We used 1-Pearson correlation coefficients as the cell-to-cell distance matrix as scOpen[10].

For data visualization, we performed PCA to reduce the dimensionality of the raw scCAS data or data enhanced by various methods to 50 and then used the t-SNE[70] and UMAP[71] algorithms to further reduce the dimension to two. Cells in the visualization could be colored by cell-type labels and batch indices. More detailed

mathematical equations and formulas for the aforementioned evaluation metrics are provided in Supplementary Text S1.

**Reporting summary**

Further information on research design is available in the Nature Portfolio Reporting Summary linked to this article.

## Data availability

All relevant data supporting the key findings of this study are available within the article and its Supplementary Information files. The Blood and BM0828 datasets and their corresponding bulk data can be retrieved from NCBI Gene Expression Omnibus (GEO) with accession number GSE96772. The datasets of various mouse tissues are available at https://atlas.gs.washington.edu/mouse-atac/data. The Muto dataset can be retrieved from GEO with accession number GSE151302. The PBMC dataset profiled by 10x Genomics via "Single Cell Multiome ATAC + Gene Expression Sequencing" can be downloaded at https://www.10xgenomics.com/resources/datasets/pbmc-from-a-healthy-donor-granulocytes-removed-through-cell-sorting-10-k-1-standard-2-0-0. The Mix-protocols dataset was concatenated from two mouse brain datasets profiled by different protocols, which are available at https://support.10xgenomics.com/single-cell-atac/datasets/1.1.0/atac_v1_adult_brain_fresh_5k and in GEO with accession number GSE126724. UCSC Genome Browser and UniProt database of protein are used in this study. Source data are provided with this paper.

## Code availability

The MIT-licensed scCASE software, including detailed documents and tutorials, is freely available on GitHub (https://github.com/BioX-NKU/scCASE). All codes for reproducing the analysis are available at Zenodo[72].

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

## Acknowledgements
This work was supported by the National Natural Science Foundation of China grant no. 62203236 (S.C.), the Young Elite Scientists Sponsorship Program by CAST grant no. 2023QNRC001 (S.C.), and the Fundamental Research Funds for the Central Universities grant no. Nankai University 63231137 (S.C.).

## Author contributions
S.C. conceived and supervised the project. S.T., R.W. and S.C. designed, implemented, and validated scCASE. X.C., S.J.L. and S.Y.L. helped analyzing the results. S.T., S.C., X.C. and X.H. wrote the manuscript with inputs from all the authors.

## Competing interests
The authors declare no competing interests.
