## [Peer Review File · Nature Communications]

scCASE: Accurate and interpretable enhancement for single-cell chromatin accessibility sequencing dataReviewer #1 (Remarks to the Author):

The authors developed scCASE, an NMF-based approach for sc-ATAC-seq denoising by leveraging information from nearby cells. They evaluated the performance of scCASE across multiple species and tissues for denoising, embedding, detecting differential peaks, and robustness. Also, they expanded the method to allow incorporation of external reference. Using NMF to model scATAC has been proposed before (scOpen), here the novel contribution is using cell-cell similarity matrix to improve imputation, and an extension to use external reference. Overall, the writing is concise and clear. However, I think several key issues need to be addressed to support their claims: 1) how model deals with over-smoothing and sequencing depth; 2) scalability; 3) detailed description on Methods section; 4) more benchmark and differential peak detection and robustness to batch. See more comments below.

Major concerns:

1. Question on over-smoothing. As a method that explicitly uses of cell-cell similarity information for imputation. It is important that the authors study potential issue of over-smoothing. As pointed out in MAGIC (PMID: 29961576) and in molecular cross validation (<https://doi.org/10.1101/786269>), imputation can result in removal of true cell-cell heterogeneity signals. In the model, Z matrix codes for cell-cell similarity and is regularized by H and lambda. Do initialization and hyperparameter choices affect over-smoothing?

2. Question on sequencing depth. Correcting for sequencing depth is one of the most important goals in denoising scATAC data, since it is more sparse than scRNA and regular TPM don't work. The authors mentioned that they "use TF-IDF transformation to normalize the sequencing depth" (Line 395). However, TF-IDF doesn't fully correct for sequencing depth. In fact, SVD after TF-IDF often reveal PC1 highly correlates with sequencing depth (PMID: 30078704). Since the scCASE doesn't model sequencing depth explicitly, is it properly accounted for?

3. Scalability (memory, run-time). The cell-cell similarity matrix and sampling matrix could be a memory bottleneck for scalability. As more scATAC atlas resources are generated, scalability is a key consideration when choosing analysis method and pipeline. The largest dataset they run scCASE on has 11k cells. Would it be possible to scale scCASE to modern atlas dataset with > 1 million cells (sci-ATAC)?

4. Better description in the Methods section. While scCASE is well explained in the Methods section, implementation details of other scATAC methods are not described. Also, implementation details and hyperparameter are not described for SNPsea, LDSC, scABC, chromVAR etc.

5. Clustering cells with scCASE. The authors described several approaches for cell clustering with scCASE. 1) clustering on PCA on the enhanced matrix (line 144); 2) clustering on H matrix (Fig. 4a); 3) scABC on the enhanced matrix. (line 263). It is not clear in each section, which approach is used and why.

For clustering evaluation, the mouse sci-ATAC-seq dataset cell type annotation are determined by latent semantic indexing followed by Louvain. Methods that show better consistency with these labels aren't necessarily better at capture true biological heterogeneity. The authors could consider additional FACS-sorted sc-ATAC datasets, or multiome dataset for consistency b/w RNA and ATAC.

6. evaluation of scCASE-detected differential peaks. The authors used scCASE to identify differential peaks, and showed that they are biologically relevant (by SNPsea, LDSC, chromVAR etc). However, they didn't compare to any baseline here. What if someone use episcanpy to find differential accessible peaks, would they achieve the similar performance.

7. batch correction evaluation (Line 320). The claim that "indicating that scCASE can also effectively enhance data with batch effects for better downstream analysis" is vague. No explicit batch correction is done here for either scCASE or other methods being compared to. If the authors want to evaluate on how well the method can do batch correction. They should consider looking at both preservation of biological variation, and batch mixing. (e.g. PMID: 34949812).

Minor concerns:

Line 170: how are the cell embeddings generated? Implementation details and hyperparameters for each method?

Fig. 4a: Are the column / row orders by hierarchical clustering? Or manually ordered. Are all peaks input to scCASE, or just differential peaks? If only differential peaks are used as inputs, isn't the analysis circular?

Line 240: there is no description of how SNPsea analysis is performed and how to interpret the results. Where does the phenotype data come from? Are SNPs called on the scATAC dataset?

Line 249: LDSC implementation details not described. What data and phenotypes are used as input?

Line 261: "we identified 1000 specific peaks ... based on the enhanced matrix". Why does the authors choose to do clustering based using enhanced matrix + scABC method? How about the cell embeddings learn by scCASE? Are the learnt cell embeddings useful for clustering?

Line 263: scABC implementation not described.

Line 263: chromVAR implementation not described. What motif base is used?

Line 304: "We randomly subsampled ...". Consider also scale up the dataset size to test memory and runtime limitation of the model on large datasets.

Reviewer #2 (Remarks to the Author):

Tang et al present scCASE, a new computational method for modeling single-cell chromatin accessibility data that enhances data analyses via non-negative matrix factorization. Through a range of benchmarks on real and simulated data, the authors argue that scCASE has superior performance.

Major concerns:

- There are now ~dozens of methods that perform similar analyses. The authors provide benchmarking that shows overall metrics like ARI but provide very limited biological intuition for why their methods outperforms any state of the art workflows.

- If this NMF workflow has a meaningful improvement in learning biological signal over a more typical PCA / LSI implementation, can the authors unpack this?

- Most of the benchmarking is done on datasets with limited cell numbers (e.g. profiling via the fludigm c1) but contemporary datasets scale to 100,000+ cells. More benchmarking / analyses on large datasets profiled via 10x scATAC-seq is required to understand scCASE's utility with modern datasets.

- > related to this, what are the considerations for scCASE's computational efficiency as datasets get larger?

- If I understand correctly, a 1% threshold for the counts matrix has to be applied to run scCASE-- this seems problematic as it would discard thousands of peaks. What's the impact of this? Is this required for computational efficiency? A more careful analyses of the upstream QC is required.

- The GitHub provides minimal working examples of the code but lags behind the more interactive vignettes of other tools (e.g. SCALE, Seurat/Signac, etc.). For adoption of the workflow, more detailed step-by-step instructions explaining the functions called is required. Further, cleaner integration with tools like scanpy and Signac would allow for much greater adoption.

Reviewer #3 (Remarks to the Author):

Single-cell chromatin accessibility sequencing provides rich data for analyzing epigenetic information and gene regulation mechanisms at the single-cell level. However, due to the sparsity and dropout events of single-cell sequencing data, how to accurately and efficiently obtain accessibility peak information has become a top priority in analyzing and applying single-cell chromatin accessibility sequencing data. In this manuscript, the authors applied non-negative matrix factorization to enhance single-cell chromatin accessibility sequencing (scCAS) data, proposing scCASE and scCASER that integrate external reference data. Compared to many published methods, it significantly enhances scCAS data, improves the credibility and interpretability of accessibility peaks, and maintains good robustness when facing different data scales and batches. This will facilitate the effective use of scCAS data and enhance its interpretability.

However, several concerns require the author's response and explanation when revising the manuscript:

1. scCASE and scCASER are both constructed based on the NMF method. What are the advantages of the number of CPU cores, GPU cores, and processing time required to enhance scCAS data of the same scale compared to other algorithms used as comparative objects in other manuscripts? I did not see this part of the data in the manuscript.
2. PCA and SVD are also commonly used high-dimensional data dimensionality reduction methods. What are the advantages of enhancing NMF in scCAS data compared to these two methods?
3. Figure 4a and Supplementary Figure 7a in the manuscript are the same. Is it a mistake or why show it two times?
4. The results shown in Figure 4a and Supplementary Figure 7 in the manuscript indicate that scCASE can effectively reveal biological insights on cell type specificity, but I have not seen corresponding results from other methods, which seem to fail to effectively demonstrate that scCASE has more advantages compared to other methods. How many overlapping peaks of cell type specificity can be obtained by different methods for the same data source?
5. As a manuscript that mainly contributes computational tools and analyses, it's better to provide the immediate data and source code for readers to reproduce the results presented in the manuscript.

To the reviewers,

We want to further thank the three reviewers for their time reviewing and their constructive comments on our manuscript entitled “Accurate and interpretable enhancement for single-cell chromatin accessibility sequencing data with scCAGE”. We have thoroughly revised and enhanced our manuscript based on their comments. In addition to the minor changes, we have made the following six major modifications in our revised manuscript, with the modifications highlighted in red. These changes are summarized in Figs. 2-5, Supplementary Texts S1, S3-7, Supplementary Figs. S5-11, S15-30, Supplementary Table S1, as well as the main text and Methods section.

1) We validated the ability of scCAGE to correct the sequencing depth, and we have designed an extension for scCAGE, provided as an optional feature of the model, that improves the normalization of sequencing depth (Fig. 2, Supplementary Text S3, and Supplementary Figs. S7, S8). Moreover, we conducted comparative experiments to demonstrate the versatility of scCAGE across datasets with complex batch effects, and we introduced another extension to scCAGE, which significantly enhances the analysis of data with batch effects (Fig. 5, Supplementary Text S5, and Supplementary Fig. S21).

2) We have updated scCAGE to achieve higher computational efficiency (Figs. R9, R20, R22) and conducted a comprehensive benchmark of the run-time and memory usage of scCAGE and baseline methods (Supplementary Fig. S29). The updated scCAGE showed a certain advantage in terms of run-time and memory usage, demonstrating the scalability on larger datasets.

3) In Section “scCAGE enhances scCAS data for better cellular heterogeneity characterization”, we validated the superior performance of scCAGE on two new datasets in which cell type annotation labels are derived from paired scRNA-seq data (Supplementary Fig. S5). We also confirmed the potential of cell embeddings learned by scCAGE in clustering. Cell embeddings of scCAGE obtained the overall best clustering performance across multiple datasets (Supplementary Fig. S6).

4) We compared the scCAGE-detected cell type-specific peaks to the differentially accessible peaks identified by EpiScanpy on raw data and the data enhanced by baseline methods. The cell type-specific peaks identified by scCAGE revealed more meaningful biological insights than baseline methods (Supplementary Text S4, Supplementary Figs. S15-20).

5) In Methods section, we additionally provided detailed descriptions of the implementation of baseline methods and the implementation of downstream analyses. Besides, we investigated the impact of initialization and hyperparameters on over-smoothing (Supplementary Figs. S26-28), and the impact of different peak filtering strategies (Supplementary Text S6, Supplementary Fig. S23).

6) We have modified the code of scCASE to make it more user-friendly and provided more detailed tutorials, such as the tutorial on the integration with EpiScanpy. We have also published all the code and immediate data used in this manuscript on Zenodo, ensuring that readers can reproduce all the results.

Next, we will clarify the comments raised by the reviewers point-by-point, with the reviewers' points in black and our responses in blue.

REVIEWER #1

The authors developed scCASE, an NMF-based approach for sc-ATAC-seq denoising by leveraging information from nearby cells. They evaluated the performance of scCASE across multiple species and tissues for denoising, embedding, detecting differential peaks, and robustness. Also, they expanded the method to allow incorporation of external reference. Using NMF to model scATAC has been proposed before (scOpen), here the novel contribution is using cell-cell similarity matrix to improve imputation, and an extension to use external reference. Overall, the writing is concise and clear. However, I think several key issues need to be addressed to support their claims: 1) how model deals with over-smoothing and sequencing depth; 2) scalability; 3) detailed description on Methods section; 4) more benchmark and differential peak detection and robustness to batch. See more comments below.

RESPONSE

We deeply appreciate the reviewer for the affirmation of our work and the enthusiastic assessment as well as thoughtful comments which helped us to improve the manuscript. We have carefully considered all the issues, responded to the raised comments point-by-point, and made revisions in the corresponding section of the manuscript.

COMMENT 1

1. Question on over-smoothing. As a method that explicitly uses of cell-cell similarity information for imputation. It is important that the authors study potential issue of over-smoothing. As pointed out in MAGIC (PMID: 29961576) and in molecular cross validation (<https://doi.org/10.1101/786269>), imputation can result in removal of true cell-cell heterogeneity signals. In the model, Z matrix codes for cell-cell similarity and is regularized by H and lambda. Do initialization and hyperparameter choices affect over-smoothing?

RESPONSE TO COMMENT 1

We express our gratitude for the reviewer's valuable insights, the recommended papers are highly enlightening for our research endeavors. We agree with the reviewer's perspective that averaging the chromatin accessibility status of each cell with that of similar cells may result in over-smoothing. In the initial manuscript, we did not study the potential issue of over-smoothing, which may result in users running scCASE with inappropriate parameters, leading to the elimination of true cell-cell heterogeneity signals in the enhanced data. We appreciate the reviewer for highlighting this potential issue. In the revised manuscript, we have investigated the possible problem of over-smoothing.

In addressing the concerns of the reviewer and potential users regarding over-smoothing, we have undertaken several efforts in this revision. First, we reviewed extensive literature to comprehend the definition of over-smoothing, relevant evaluation standards, and associated metrics. Secondly, we summarized the descriptions from the literature on over-smoothing and proposed metrics to assess the extent of over-smoothing. Lastly, we performed scCASE with varied initialization and hyperparameters to investigate the smoothing tendencies under different settings.

To comprehend how to assess the extent of over-smoothing, we carefully studied the descriptions provided by MAGIC¹ and MCV². MAGIC demonstrated the model's robustness across diverse hyperparameter selections and elucidated that selecting excessively large values for parameters such as the ka (distance to its k^{th} nearest neighbor) and t (diffusion time) of MAGIC could lead to over-smoothing. MCV mentioned that each of the denoising methods has parameters that control the trade-off between removing noise and blurring the biological signal. To illustrate that inappropriate parameter selection can lead to over-smoothing and impact the downstream analyses, MCV provided gene-cell heatmaps which depicted the results of denoising using principal component analysis (PCA) with different numbers of principal components, illustrating that over-smoothing with inappropriate parameters (number of principal components equals three) leads to the elimination of cellular heterogeneity between different cell types. Additionally, the study of MCV showed relationship plots between genes, indicating that over-smoothing of data can result in a pronounced correlation between genes that exhibited no correlation in the raw data.

In addition to the above two reference papers suggested by the reviewer, we conducted an extensive literature review on the issue of over-smoothing in single-cell data imputation. Tran et al. employed gene-cell heatmaps and t-distributed stochastic neighbor embedding (t-SNE) visualizations (Fig. S16 of Tran et al.) to assess over-smoothing in the scRNA-seq data imputed by different methods³. Their findings indicated that over-smoothing leads to the elimination of heterogeneity between different cell types, merging all the cells in different types together, and rendering them indistinguishable in heatmap or t-SNE visualizations. Tjärnberg et al. suggested that over-smoothing might lead to high correlations between cells in different subpopulations⁴. They substantiated that their method did not induce over-smoothing by demonstrating that in the data enhanced by their method, cells of the same type are strongly correlated, whereas cells of different types are weakly correlated when plotted as a heatmap (Fig. S1 of Tjärnberg et al.). Furthermore, SCDD elucidated that over-smoothing detrimentally affects clustering metrics, such as the adjusted Rand index (ARI), resulting in poorer performance compared to the raw data (Section “Effects of clustering” of SCDD)⁵. In conclusion, adequate data enhancement can preserve heterogeneity between different cell types and eliminate data noise. However

over-smoothing results in cells with substantial differences exhibiting an excessive degree of similarity, leading to entirely uniform gene expression/chromatin accessibility and the elimination of cellular heterogeneity.

Although the studies on over-smoothing in scRNA-seq imputation methods have provided valuable insights for our research. There are certain limitations in the existing literature concerning the investigation of over-smoothing. On the one hand, despite the widespread use of methodologies such as heatmaps, t-SNE/UMAP (uniform manifold approximation and projection) visualizations, and clustering metrics in evaluating over-smoothing, there is an absence of dedicated metrics specifically designed for characterizing over-smoothing. This hinders our ability to undertake a quantitative study on over-smoothing. On the other hand, these studies predominantly focus on scRNA-seq data, and our search using keywords "over-smooth" and "Single-cell assay for transposase-accessible chromatin by sequencing (scATAC-seq)/single-cell chromatin accessibility sequencing (scCAS)" on Google Scholar did not yield relevant research on over-smoothing in the field of scATAC-seq/scCAS data imputation methods. There are substantial differences between scRNA-seq and scCAS data. For instance, the sparsity and binary property of scCAS render the computation of correlations between two cells less meaningful. This impedes our ability to assess whether over-smoothing has occurred by evaluating the correlation in chromatin accessibility before and after imputation (as done in MAGIC and MCV).

To address the aforementioned issues, in this revision, we have devised several simple metrics to assess the degree of over-smoothing in the enhanced data and evaluate the effectiveness of enhancement process. We posit that the data affected by over-smoothing would be an exaggerated similarity between certain distinct cell types. In contrast, well-smoothed data would lead to closer distances between cells of the same type, whereas distances between a cell and another cell from different types would be greater than those between two cells of the same type. Based on this, let $d(x_1, x_2)$ represent the distance between cells x_1 and x_2 (assumed to be the Euclidean distance in the original dimensions). For a given cell x , let a be the average distance between x and other cells of the same type with x , and let b be the average distance between x and cells of the nearest different cell type. We can compute the averages \bar{a} and \bar{b} for a and b of all cells in a dataset, and calculate the average distance \bar{d} between any two cells in the dataset. We define the over-smoothing score as \bar{b}/\bar{d} and the under-smoothing score is $1 - \bar{a}/\bar{d}$. For the over-smoothing score, a low score implies that each cell has a close distance to a different cell type, indicating excessive smoothing that eliminates cellular heterogeneity. Thus, a smaller over-smoothing score is associated with a higher degree of over-smoothing. Regarding the under-smoothing score, a lower value suggests that the distances between cells of the same type have not become closer after enhancement. Under-

smoothing scores are scaled and larger scores are associated with better performance. To better quantify the enhancement effect, we compute the harmonic mean of the over-smoothing score and the under-smoothing score, referred to as the smoothing score.

$$\text{smoothing score} = \frac{2 \times \text{over smoothing score} \times \text{under smoothing score}}{\text{over smoothing score} + \text{under smoothing score}}$$

We also computed adjusted Rand index (ARI), adjusted mutual information (AMI), Fowlkes-Mallows index (FMI), and silhouette score. In addition to the aforementioned metrics, we generated UMAP visualizations and peak-cell heatmaps for different parameters (for clarity, the heatmap only depicts the top 10 cell type-specific peaks identified by EpiScanpy in the raw data). This comprehensive set of assessments aims to better assess the presence of over-smoothing.

We performed scCASE with varied initialization and hyperparameters to investigate the smoothing tendencies under different settings, taking a simulated dataset as an example, featuring ten cell types, each consisting of 500 cells. First, for model initialization, in scCASE, we initialized the similarity matrix \mathbf{Z} using the Jaccard correlation coefficient and the cell

Fig. R1. Performance of scCASE with different initialization. **a**, The enhancement performance of different initialization. **b**, UMAP visualization of data enhanced by scCASE with random initialization of \mathbf{W} and \mathbf{H} . **c**, Cell-peak heatmap of data enhanced by scCASE with random initialization of \mathbf{W} and \mathbf{H} .

embedding \mathbf{H} using the result of non-negative matrix factorization. A random initialization of \mathbf{Z} or \mathbf{H} can lead to mutual influence, causing the model to converge to unexpected results. As \mathbf{Z} is constrained by both \mathbf{H} and lambda, random initialization of \mathbf{H} would severely affect the matrix \mathbf{Z} , leading to a totally unsatisfactory outcome (Fig. R1). The random initialization of \mathbf{Z} is less impactful, and the model will still converge, although it may affect the model to some extent (Fig. R1), leading to a worse performance of enhancement results in lower scores for under-smoothing and smoothing, but the high over-smoothing score suggests that the random initialization of \mathbf{Z} does not lead to over-smoothing. The lower clustering metrics showed the random initialization of \mathbf{Z} also led to a deterioration in clustering performance.

Secondly, we explored the results with different values of lambda. The default value of lambda in the model is 10^6 , and we ran scCASE with varying lambda values within the range

Fig. R2. Performance of scCASE with varying hyperparameter Lambda. **a**, The enhancement performance of scCASE with varying hyperparameter Lambda. **b**, UMAP visualization of data enhanced by scCASE with varying hyperparameter Lambda. **c**, Cell-peak heatmap of the data enhanced by scCASE with varying hyperparameter Lambda.

of 10^5 to 10^8 to investigate the impact of lambda on over-smoothing. As lambda varies, different metrics remain stable with few changes (Fig. R2). This indicates the high robustness of scCASE to the choice of lambda, suggesting that within a certain range, the choice of lambda does not lead to over-smoothing.

Finally, we validated the impact of different values of parameter K , namely the dimensions of non-negative matrix factorization. The results indicate that when K is small ($K < 7$), it can be observed that at lower dimensions, the model struggles to capture differences in the data effectively, leading to the elimination of heterogeneity between different cell types and over-smoothing of the data (Figs. R3-R5). As K gradually increases, the degree of over-smoothing eases. However, though not lead to over-smoothing, large values of K ($K > 20$) may introduce excessive noise, making the model learning more challenging and resulting in a lower under-smoothing score (Figs. R3-R5).

Fig. R3. Performance of scCASE with varying hyperparameter K . **a**, The bar plot of over-smoothing score, under-smoothing score and smoothing score of scCASE with varying hyperparameter K . **b**, The bar plot of ARI, AMI, FMI and Silhouette score of scCASE with varying hyperparameter K .

Fig. R4. UMAP visualization of the data enhanced by scCASE with varying hyperparameter K .

Fig. R5. Cell-peak heatmap of the data enhanced by scCASE with varying hyperparameter K .

In summary, in this section, we discussed whether the initialization and hyperparameter choices of scCASE would lead to over-smoothing. The results indicate that the initialization and choice of lambda do not result in over-smoothing in scCASE. Lower values of K ($K < 7$) may lead to the model's inability to capture heterogeneities between cells, thus causing a certain degree of over-smoothing issue. We have added this discussion in Section “Initialization and parameter selection of scCASE” in the revised manuscript (Supplementary Figs. S24-S28). Thank you again for the valuable comment.

COMMENT 2

2. Question on sequencing depth. Correcting for sequencing depth is one of the most important goals in denoising scATAC data, since it is more sparse than scRNA and regular TPM don't work. The authors mentioned that they “use TF-IDF transformation to normalize the sequencing depth“ (Line 395). However, TF-IDF doesn't fully correct for sequencing depth. In fact, SVD after TF-IDF often reveal PC1 highly correlates with sequencing depth (PMID: 30078704). Since the scCASE doesn't model sequencing depth explicitly, is it properly accounted for?

RESPONSE TO COMMENT 2

We express our gratitude for this insightful comment. We agree with the reviewer's perspective, correcting for sequencing depth is one of the most important goals in scATAC-seq data analysis. To assess the validity of scCASE in correcting for sequencing depth, we conducted the following experiments.

Following a workflow similar to Cusanovich DA et al.⁶, we applied TF-IDF transformation to the raw count matrix and employed SVD to reduce the model dimensions to 10. As pointed out by the reviewer, we indeed observed a significant correlation between PC1 and sequencing depth (Figs. R6a, d, R7a, d). For instance, in the Blood dataset, the correlation coefficient between PC1 and sequencing depth exceeds 0.8 (Fig. R6d). This is also evident in UMAP visualization, where sequencing depth largely determines the positions of cells in the low-dimensional representation (Fig. R6a). In the LungA dataset, we observed similar results, the correlation coefficient between PC1 and sequencing depth exceeds 0.7 (Fig. R7d), and the impact of sequencing depth on low-dimensional representation can also be observed in UMAP visualization (Fig. R7a). This indicates that the raw data combined with TF-IDF does not effectively normalize the sequencing depth, which is consistent with the reviewer's claim.

Fig. R6. Impact of sequencing depth on raw data and the data enhanced by scCASE on the Blood dataset. UMAP visualization of **a**, raw data, **b**, the data enhanced by scCASE and **c**, the data enhanced by the extended scCASE. **d-f**, Correlation coefficient between sequencing depth and SVD components of raw data, the data enhanced by scCASE and the extended scCASE, respectively.

For the data enhanced by scCAGE, we conducted a similar process, and the results indicate that, despite the absence of explicit modeling for sequencing depth, the observed correlation was somewhat attenuated. In the Blood dataset, the correlation coefficient between PC1 and sequencing depth has been reduced by 46.5%, while in the LungA dataset, it has been reduced by 19.5% (Figs. R6d, e, R7d, e). Given that scCAGE takes into consideration the similarity between cells during modeling and similar cells generally exhibit comparable chromatin accessibility patterns, a particular cell has numerous similar cells with varying sequencing depths. Representing read counts as the weighted average of multiple similar cells corrected the sequencing depth at the same time. Consequently, in the scCAGE-enhanced data, the influence of sequencing depth is mitigated to a certain extent.

To improve the capability of scCAGE in further mitigating the impact of sequencing depth, we extended the scCAGE model and set it as optional for users. In this extended model, we explicitly incorporated sequencing depth for modeling by introducing the sequencing depth matrix $\mathbf{P} \in \mathbb{R}^{n \times n}$. The elements on the main diagonal of this matrix represent the sequencing depth of the cells. This matrix is utilized to weight the similarity matrix \mathbf{Z} , aiming to describe the correlation between the similarity matrix and sequencing depth. This explicit consideration

Fig. R7. Impact of sequencing depth on raw data and the data enhanced by scCAGE on the LungA dataset. UMAP visualization of **a**, raw data, **b**, the data enhanced by scCAGE and **c**, the data enhanced by the extended scCAGE. **d-f**, Correlation coefficient between sequencing depth and SVD components of raw data, the data enhanced by scCAGE and the extended scCAGE, respectively.

serves to minimize the influence of sequencing depth on the enhanced data. The specific formulation of this extended version of scCASE is as follows:

$$\min_{\mathbf{W}, \mathbf{H}, \mathbf{Z} \geq 0} F = \|\mathbf{X}(\mathbf{Z} \circ \mathbf{R}) - \mathbf{W}\mathbf{H}\|_F^2 + \lambda \|\mathbf{Z} - \mathbf{P}(\mathbf{H}^T \mathbf{H})\|_F^2 + \gamma_1 \|\mathbf{W}\|_F^2 + \gamma_2 \|\mathbf{H}\|_F^2$$

We can convert the loss function from a norm form to a trace form as follows, making it easier to compute the gradient. After obtaining the partial derivatives of F with respect to \mathbf{W} , \mathbf{H} , and \mathbf{Z} , we use gradient descent to optimize the model.

$$F = tr((\mathbf{Z} \circ \mathbf{R})^T \mathbf{X}^T \mathbf{X} (\mathbf{Z} \circ \mathbf{R}) - \mathbf{H}^T \mathbf{W}^T \mathbf{X} (\mathbf{Z} \circ \mathbf{R}) - (\mathbf{Z} \circ \mathbf{R})^T \mathbf{X}^T \mathbf{W} \mathbf{H} + \mathbf{H}^T \mathbf{W}^T \mathbf{W} \mathbf{H}) \\ + \lambda tr(\mathbf{Z}^T \mathbf{Z} - \mathbf{P}(\mathbf{H}^T \mathbf{H}) \mathbf{Z} - \mathbf{Z}^T (\mathbf{P}(\mathbf{H}^T \mathbf{H}))) + (\mathbf{P}(\mathbf{H}^T \mathbf{H})) (\mathbf{P}(\mathbf{H}^T \mathbf{H})) \\ + \gamma_1 tr(\mathbf{W}^T \mathbf{W}) + \gamma_2 tr(\mathbf{H}^T \mathbf{H})$$

$$\frac{\partial F}{\partial \mathbf{W}} = -2\mathbf{X}(\mathbf{Z} \circ \mathbf{R})\mathbf{H}^T + 2\mathbf{W}\mathbf{H}\mathbf{H}^T + 2\gamma_1 \mathbf{W}$$

$$\frac{\partial F}{\partial \mathbf{H}} = -2\mathbf{W}^T \mathbf{X}(\mathbf{Z} \circ \mathbf{R}) + 2\mathbf{W}^T \mathbf{W} \mathbf{H} - 2\lambda \mathbf{H}(\mathbf{Z} \mathbf{P} + (\mathbf{Z} \mathbf{P})^T) + 2\lambda \mathbf{H}(\mathbf{H}^T \mathbf{H} \mathbf{P}^T \mathbf{P} + \mathbf{P}^T \mathbf{P} \mathbf{H}^T \mathbf{H}) + 2\gamma_2 \mathbf{H}$$

$$\frac{\partial F}{\partial \mathbf{Z}} = 2(\mathbf{X}^T \mathbf{X} (\mathbf{Z} \circ \mathbf{R})) \circ \mathbf{R} - 2(\mathbf{X}^T \mathbf{W} \mathbf{H}) \circ \mathbf{R} + 2\lambda \mathbf{Z} - 2\lambda \mathbf{P} \mathbf{H}^T \mathbf{H}$$

We validated the efficacy of this model in effectively correcting for sequencing depth. Using the same evaluation metrics, we applied SVD to the enhanced data and calculated the correlation coefficients between each component and sequencing depth. In the data enhanced by the extended scCASE, it is hard to observe a correlation between individual principal components and the sequencing depth (Figs. R6c, f, R7c, f). Additionally, we conducted a clustering performance assessment for this enhanced method. The results suggest that the extension achieved enhancement performance similar to the original version, while successfully mitigating the impact of sequencing depth (Figs. R6c, R7c). We have made this extension available as an optional variant of scCASE and provided a tutorial on GitHub.

In summary, despite not explicitly modeling sequencing depth, scCASE is still capable of correcting the sequencing depth to some extent. Additionally, we have designed an extension for scCASE, provided as an optional variant of the model, that improves the normalization of sequencing depth. We have included the results and discussion in Section “scCASE enhances scCAS data for better cellular heterogeneity characterization” (Fig. 2, Supplementary Figs. S7, S8) and Supplementary Text S3 in the revised manuscript.

COMMENT 3

3. Scalability (memory, run-time). The cell-cell similarity matrix and sampling matrix could be a memory bottleneck for scalability. As more scATAC atlas resources are generated, scalability is a key consideration when choosing analysis method and pipeline. The largest dataset they run scCASE on

has 11k cells. Would it be possible to scale scCASE to modern atlas dataset with > 1 million cells (sci-ATAC)?

RESPONSE TO COMMENT 3

We appreciate the reviewer's keen insights, and we concur with the view that the cell-cell similarity matrix and sampling matrix could pose a memory bottleneck for scalability. We apologize for not including comparisons related to run-time and memory usage in the manuscript. In this revision, we have provided a comparison of run-time and peak memory usage across multiple datasets. The hardware configuration utilized in our experiments is outlined below: 500GB memory, Intel Xeon Gold 6348 CPU @ 2.60GHz with 112 cores, Ubuntu 22.04.2 LTS. For SCALE and scBasset, two deep learning-based methods, we used NVIDIA A40 with 48GB memory.

Firstly, we evaluated the computational efficiency of the original scCASE. The run-time and peak memory usage of the original scCASE code on different datasets are provided in Fig. R8. We recognize that the initial version of scCASE did not exhibit an advantage of computational efficiency over other methods. In response to the reviewer's emphasis on run-time and peak memory usage considerations, we have performed a comprehensive update to the implementation of scCASE. The updated version incorporates several optimizations to speed up scCASE. Notable optimizations include: 1) Jaccard similarity calculation: we updated the Jaccard similarity calculation method by utilizing sparse matrix operations instead of the original Jaccard calculation method provided by Scipy. 2) Data storage types: we adjusted the precision of data storage from double-precision floating-point to single-precision floating-point. This approach reduced memory usage and did not impact experimental results. 3) Matrix operation optimization: we refactored the matrix computation process in the code by employing

Fig. R8. Run-time and peak memory usage of original implementation of scCASE and the updated implementation of scCASE. a, Run-time of original implementation of scCASE and the updated implementation of scCASE. **b,** Peak memory usage of original implementation of scCASE and the updated implementation of scCASE.

Fig. R9. Run-time and peak memory usage of updated scCASE and baseline methods. a, Run-time of scCASE and baseline methods. **b,** Peak memory usage of scCASE and baseline methods.

faster matrix operation functions and reducing the use of intermediate variables to expedite the computation of iteration results and reduce memory consumption. These optimizations solely reduce run-time and memory usage and will not have any impact on the results presented in the manuscript.

A comparative analysis of the run-time and peak memory usage of scCASE before and after these optimizations across various datasets is presented in Fig. R8. These optimizations have led to a substantial reduction in peak memory usage and run-time consumption. Specifically, on the Blood and LungA datasets, there is an average reduction of approximately 30% in peak memory usage and an average reduction of approximately 80% in run-time. Meanwhile, on the larger Muto dataset, the improved scCASE required only half the original peak memory usage and one-tenth of the original run-time (Fig. R8). This indicates that the improvements are particularly pronounced when dealing with larger datasets. Furthermore, these improvements enable scCASE to successfully enhance datasets with 100k cells, which is too large for the original scCASE to run in the limited hardware.

We further conducted extensive testing on diverse datasets, expanding the maximum number of cells to 100k. In our evaluations, we compared the optimized scCASE with other baseline methods in terms of run-time (Fig. R9a) and peak memory usage (Fig. R9b). The updated scCASE consistently exhibited commendable performance. In terms of run-time, scCASE exhibits significant advantages compared to other methods, especially on smaller datasets such as BM0828, Blood, and LungA, where scCASE can operate several times faster than baseline methods (Fig. R9a). Even on larger datasets, scCASE still maintains a notable speed advantage (Fig. R9a). In terms of peak memory usage, SCALE and scBasset, two GPU-based methods, make more usage of GPU memory, and thus typically require less memory usage than the methods that utilize CPUs. scCASE demonstrates a certain advantage in peak

memory usage on smaller datasets (BM0828, Blood, and LungA) (Fig. R9b). Although the peak memory usage of scCASE increases in larger datasets, its memory usage is still comparable to that of scOpen, the state-of-the-art scCAS data enhancement method (Fig. R9b). Moreover, the memory usage of scCASE growth remains manageable. On two larger datasets of Muto and Simulated, which have a similar number of peaks but a fivefold increase in the number of cells (from 20k to 100k), the peak memory usage of scCASE increased by 7.67 times, while that of scOpen increased by 8.62 times. Note that scBFA is unable to run on datasets with 100k cells due to out-of-memory errors.

In conclusion, we have improved the computational efficiency of scCASE and compared the run-time and peak memory usage of scCASE with that of baseline methods. scCASE demonstrates a certain advantage in terms of run-time and memory usage compared to other methods. The memory usage of scCASE grows manageable, highlighting its scalability on larger datasets. We have included the results in Section “Run-time and memory usage of scCASE” in the revised manuscript (Supplementary Fig. S29).

COMMENT 4

4. Better description in the Methods section. While scCASE is well explained in the Methods section, implementation details of other scATAC methods are not described. Also, implementation details and hyperparameter are not described for SNPsea, LDSC, scABC, chromVAR etc.

RESPONSE TO COMMENT 4

We sincerely apologize for not providing detailed descriptions of the implementation details of other methods in the first version of our manuscript. Firstly, following the reviewer's suggestion, we have provided detailed descriptions for running other scATAC-seq methods in the Methods section of the revised manuscript. Here we review this part for your convenience:

scBFA: scBFA is a detection-based model to remove technical variation in scRNA-seq and scATAC-seq data, available at <https://github.com/quon-titative-biology/scBFA>⁷. We utilized the raw count matrices as input and performed scBFA using their default parameters. We executed scBFA following the same benchmarking procedure as in scOpen. (<https://github.com/CostaLab/scopen-reproducibility/blob/main/scripts/Imputation/scBFA.R>).

SCALE: SCALE integrates both the variational auto-encoder (VAE) and the Gaussian mixture model (GMM) to characterize the distribution of scATAC-seq data⁸. We obtained the SCALE program from <https://github.com/jsxlei/SCALE> and used the raw count matrices as

input. When executing the program, we set the option "impute" as TRUE to obtain the imputed data, while keeping other parameters at their default settings.

scBasset: scBasset is a sequence-based convolutional neural network method to model scATAC data and predict chromatin accessibility⁹. The program and training tutorial of scBasset can be downloaded from <https://github.com/calico/scBasset>. We trained scBasset with default parameters using the raw count matrices and peaks as the input. Genome fasta file used in scBasset can be downloaded from <https://hgdownload.soe.ucsc.edu/downloads.html>. After obtaining the trained model, we referred to their tutorial to obtain the enhanced data (https://github.com/calico/scBasset/blob/main/examples/PBMC_multiome/evaluate.ipynb).

scOpen: scOpen is a scATAC-seq imputation method based on regularized non-negative matrix factorization¹⁰. The raw scATAC count matrix serves as the input for scOpen, and the output is an imputed matrix. We followed the tutorial and examples of scOpen provided at <https://github.com/CostaLab/scopen> and executed it with default parameters.

Furthermore, we have supplemented implementation details and hyperparameters in the revised manuscript for other analyses, such as SNPsea, LDSC, scABC, and chromVAR, which have been thoroughly described in Responses to Comments #8, #10, #11, #13, and #14 to Reviewer #1. All the code used to implement the above methods and other analyses are provided on Zenodo¹¹ for reproduction. In the revised manuscript, we have incorporated the implementation details in Sections "Implementation details of baseline methods" and "Implementation details of downstream analyses".

COMMENT 5

5. Clustering cells with scCASE. The authors described several approaches for cell clustering with scCASE. 1) clustering on PCA on the enhanced matrix (line 144); 2) clustering on H matrix (Fig. 4a); 3) scABC on the enhanced matrix. (line 263). It is not clear in each section, which approach is used and why.

For clustering evaluation, the mouse sci-ATAC-seq dataset cell type annotation are determined by latent semantic indexing followed by Louvain. Methods that show better consistency with these labels aren't necessarily better at capture true biological heterogeneity. The authors could consider additional FACS-sorted sc-ATAC datasets, or multiome dataset for consistency b/w RNA and ATAC.

RESPONSE TO COMMENT 5

We appreciate this meticulous comment and apologize for any lack of clarity in our manuscript. Next, we will provide a detailed explanation of the three points mentioned by the reviewer.

Actually, we employed only one clustering method, namely 1) clustering on PCA on the enhanced matrix. All clustering analyses were performed using this method. For 2), Fig. 4a did not involve clustering results; instead, it utilized the cell type annotation labels of the FACS-sorted Blood dataset. For 3), we sincerely apologize for any confusion caused by the unclear description in Line 263 that led the reviewer to misunderstand that scABC was employed for clustering. scABC was used to identify specific peaks for each cluster, aligning with the analysis workflow following RA3¹². The clustering labels were obtained through method 1). We have revised the manuscript to eliminate any potential misunderstanding for the readers.

We deeply agree with the reviewer that labels obtained through latent semantic indexing (LSI) and Louvain clustering may not necessarily accurately reflect the true cell heterogeneity. However, there is almost no real “gold” standard dataset. Although a small number of datasets with cell types identified by FACS sorting are available, these data are often biased toward certain cell types since the cell surface markers are not available for all cell types¹³. There sure will be mislabels in the available scATAC-seq data, but we believe those are random errors and shouldn’t have a systematic impact on the assessment of cell type annotation performance, since the datasets and labels have also been used in other previous studies^{12, 14-18}. Even that, to address the reviewer’s concern, we further incorporated multiome datasets of Muto and PBMC. The Muto dataset was profiled via snATAC-seq and snRNA-seq and contains human kidney cells¹⁹. The authors annotated the snRNA-seq data based on lineage-specific marker expression and then leveraged the annotated snRNA-seq dataset to predict snATAC-seq cell types with the label transfer function in Seurat²⁰. The distribution of snATAC-seq prediction scores showed that the vast majority of cells had a high prediction score and were confidently assigned to a cell type¹⁹. The PBMC dataset was profiled by 10x Genomics via “Single Cell Multiome

Fig. R10. Evaluation of clustering performance on the Muto and PBMC datasets. a, The values of different clustering metrics on the Muto dataset. **b**, The values of different clustering metrics on the PBMC dataset.

ATAC + Gene Expression Sequencing” and contains the cryopreserved human peripheral blood mononuclear cells (PBMCs) of a healthy female donor. The cell type labels are annotated using only the scRNA-seq data (<https://www.10xgenomics.com/resources/datasets/pbmc-from-a-healthy-donor-granulocytes-removed-through-cell-sorting-10-k-1-standard-2-0-0>). The Muto and PBMC datasets are widely used in previous studies^{10, 21, 22}. We included these two datasets in our benchmark and demonstrated that scCASE also exhibits excellent performance. We assessed the clustering results using ARI, AMI, FMI, and silhouette score, and the data enhanced by scCASE consistently demonstrated the best performance compared to raw data and the data enhanced by baseline methods (Fig. R10). In summary, we have verified the performance of scCASE in datasets annotated by various methods, including FACS-sorted, LSI + Louvain clustering, and multiome dataset for consistency b/w RNA and ATAC, where the cell type annotation labels of these datasets have been widely used for evaluation. The advantages demonstrated by scCASE across these datasets underscore its broad effectiveness.

In conclusion, we apologize again for any lack of clarity in our manuscript. We performed clustering using only “1) clustering on PCA on the enhanced matrix”. Additionally, we employed the multiome datasets of Muto and PBMC to validate the performance of scCASE and demonstrated the superior performance of scCASE. In the revised manuscript, we have revised the corresponding descriptions and incorporated the additional datasets in Section “scCASE enhances scCAS data for better cellular heterogeneity characterization” of the revised manuscript (Supplementary Fig. S5).

COMMENT 6

6. evaluation of scCASE-detected differential peaks. The authors used scCASE to identify differential peaks, and showed that they are biologically relevant (by SNPsea, LDSC, chromVAR etc). However, they didn’t compare to any baseline here. What if someone use episcanpy to find differential accessible peaks, would they achieve the similar performance.

RESPONSE TO COMMENT 6

We appreciate the reviewer for this insightful comment. Firstly, our description of downstream analysis aims to illustrate that scCASE is an interpretable method with the advantage of extracting biological insights for specific cell populations while enhancing scATAC-seq data. This is a unique feature that many other scCAS data enhancement methods cannot accomplish. Taking scOpen as an example, its tutorial only performs cell clustering using data enhanced by scOpen and uses EpiScanpy on the raw data to identify specific peaks. Conversely, for scCASE

Fig. R11. The overlap of cell type-specific peaks identified by scCASE and the differentially accessible peaks identified by EpiScanpy on the raw data.

we can consider a column of the projection matrix (a pattern of peaks) corresponding to the row of cell embedding with the highest activation levels in a certain cluster, investigate the pattern of peaks with relatively large coefficients, and identify the cell type-specific peaks. This illustrates that the scCASE model can learn peak accessibility patterns of cell types, providing more biological insights than other methods and demonstrating better interpretability.

Secondly, we followed the reviewer's suggestion to comprehensively compare the cell type-specific peaks identified by scCASE and the differentially accessible peaks (DAPs) identified by EpiScanpy. Taking the Blood dataset as an example again, we computed the overlap between the scCASE-identified cell type-specific top 1000 peaks and the top 1000 DAPs identified by EpiScanpy. Figure R11 illustrates a moderate degree of overlap between the cell type-specific peaks identified by scCASE and the DAPs identified by EpiScanpy. In six cell types, including CLP, CMP, LMPP, MEP, mono, and pDC, 50%-80% of the peaks are overlapped, and there is less degree of overlap in the cell types of HSC and GMP, respectively with 341 and 343 overlapping peaks. The moderate degree of overlap not only indicates the reliability of the cell type-specific peaks identified by scCASE, as they align well with a widely used tool like EpiScanpy but also highlights scCASE's ability to uncover information that is challenging to discern in the original data.

Due to the limited overlap between cell type-specific peaks identified by scCASE and the DAPs identified by EpiScanpy of HSC and GMP cells, we used these two types as an example to demonstrate how the specific peaks identified by scCASE contribute to superior biological

Fig. R12. SNPsea enrichment analysis for the cell type-specific peaks identified by scCASE and the differentially accessible peaks identified by EpiScanpy on the raw data. **a-d**, scCASE-identified cell type-specific peaks. **e-h**, DAPs identified by EpiScanpy.

insights into cellular heterogeneity. The two methods had four sets of peaks across the two cell types, including HSC-specific peaks identified by scCASE, GMP-specific peaks identified by scCASE, DAPs of HSC identified by EpiScanpy and DAPs of GMP identified by EpiScanpy. To investigate the differences in the peaks obtained by the two methods, we removed their intersection in each type, resulting in an additional four sets of peaks, including scCASE-unique HSC-specific peaks, scCASE-unique GMP-specific peaks, the unique DAPs of HSC identified by EpiScanpy, and the unique DAPs of GMP identified by EpiScanpy. Then we performed single-nucleotide polymorphisms (SNPs) enrichment analysis using SNPsea to obtain tissues explicitly affected by these peaks. Note that hematopoietic stem cells (HSCs) serve as the foundational source for immune cells, including T cells and B cells and GMP (granulocyte-macrophage progenitor) cells represent a stage in hematopoiesis and give rise to various immune cells²³⁻²⁵. HSCs and GMPs play a central role in orchestrating the generation and continuous replenishment of various immune cell types, contributing to the overall functionality of the immune system. Therefore, the HSC and GMP specificity peaks should exhibit a higher correlation with whole blood, myeloid cells, and lymphocytes. In the specific peaks obtained by scCASE, we can significantly observe this correlation (Fig. R12a-d). However, the specific peaks unearthed by EpiScanpy do not capture this correlation effectively (Fig. R12e-h). The genomic region enrichment of annotation tool (GREAT) analysis of scCASE-unique HSC-specific peaks obtained 20 pathways, comprehensively associated with functions such as immune regulation, immune cell activation, hematopoietic regulation, etc. In contrast, the unique DAPs of HSC identified by EpiScanpy consist of only four pathways, solely related to immune regulation.

Finally, we applied EpiScanpy to the data enhanced by baseline methods to identify DAPs and to compare the potential of scCASE and other baseline methods in uncovering biological insights. We first generated an upset plot for each group of peaks, and the results showed a higher intersection between EpiScanpy + raw data and scCASE, while the intersections between EpiScanpy + data enhanced by other methods were less prominent (Fig. R13). To validate whether EpiScanpy can identify DAPs with greater biological specificity from data enhanced by baseline methods, we utilized DAPs identified by EpiScanpy on the raw data, DAPs identified by EpiScanpy on the data enhanced by scOpen, and cell type-specific peaks identified by scCASE as examples, given DAPs identified by EpiScanpy + scOpen had minimal overlap with that by EpiScanpy + raw data (Fig. R13). We generated heatmaps using the raw data and different sets of peaks (Fig. R14). Figure R14a displays DAPs identified by EpiScanpy + raw data, Fig. R14b showcases DAPs identified by EpiScanpy + scOpen, and Fig. R14c presents cell type-specific peaks identified by scCASE. It was observed in Fig. R14b that when EpiScanpy was applied to the data enhanced by scOpen, although the specific peaks

Fig. R13. The overlap of cell type-specific peaks identified by scCASE and the differentially accessible peaks identified by EpiScanpy on the raw data and the data enhanced by baseline methods.

Fig. R14. Cell-peak heatmap of the raw data. a, Differentially accessible peaks identified by EpiScanpy on the raw data. **b,** Differentially accessible peaks identified by EpiScanpy on the data enhanced by scOpen. **c,** Cell type-specific peaks identified by scCASE.

Fig. R15. SNPsea analysis for the scCASE-identified cell type-specific peaks and accessible peaks identified by EpiScanpy on the raw data and the data enhanced by baseline methods. a, HSC-specific peaks. **b,** GMP-specific peaks.

obtained were indeed specific, compared to the original data, these peaks exhibited low accessibilities in each type and higher randomness. Therefore, they are not the biological cell type-specific peaks we aimed to identify. In other words, the accessibilities of the identified peaks are limited, and they are only sporadically accessible in certain cells, rather than being specific to that particular cell type, holding less biological significance. The enhancement by scOpen magnified such signals, which is not desired. Downstream analysis also confirmed that these peaks do not effectively reflect cellular heterogeneity. We implemented downstream analysis using cell type-specific peaks identified by scCASE and the DAPs identified by EpiScanpy + data enhanced by baseline methods (Figs. R15, R16). With regard to SNPs enrichment analysis, HSC-specific peaks and GMP-specific peaks identified by scCASE demonstrate better cell type-specificity compared to EpiScanpy + data enhanced by baseline methods (Fig. R15). The heritability enrichment analysis showed that the blood-related phenotypes exhibited higher associations with the HSC-specific peaks and GMP-specific peaks identified by scCASE than that of EpiScanpy + data enhanced by baseline methods (Fig. R16).

Overall, the enrichment results with the cell type-specific peaks identified by scCASE are relatively better than those identified by EpiScanpy in the raw data or data enhanced by baseline

Fig. R16. Heritability enrichment estimates for the cell type-specific peaks identified by scCASE and accessible peaks identified by EpiScanpy on the raw data and the data enhanced by baseline methods.

methods, demonstrating the biological significance of the cell type-specific peaks identified by scCASE. In the revised manuscript, we have incorporated the comparison of biological analysis in Section “scCASE intuitively reveals cell type-specific biological insights” (Supplementary Figs. S15-S20) and Supplementary Text S4. We once again appreciate the reviewer for this constructive comment.

COMMENT 7

7. batch correction evaluation (Line 320). The claim that “indicating that scCASE can also effectively enhance data with batch effects for better downstream analysis” is vague. No explicit batch correction is done here for either scCASE or other methods being compared to. If the authors want to evaluate on how well the method can do batch correction. They should consider looking at both preservation of biological variation, and batch mixing. (e.g. PMID: 34949812).

RESPONSE TO COMMENT 7

We deeply appreciate the reviewer for this knowledgeable comment, and we apologize for any ambiguity. Our intention was not to assert that scCASE can address batch effects. As a method primarily dedicated to enhancing scCAS data, we aim to preserve the authentic cell-to-cell variability existing among different cell types and impute the dropout events. It means that batch correction is not within the scope of scCASE. scCASE cannot discern whether differences arise from batch effects or inherent biological heterogeneity in the data. The section corresponding to Line 320 aims to elucidate that scCASE is effective in diverse scenarios. Regardless of high sparsity, imbalance, or the presence of severe batch effects, scCASE demonstrates robust data enhancement. We have rectified the description in this revision.

Even so, the reviewer's comment provided us with valuable insights. Consequently, we further proposed an optional extension of scCASE to model batch effects. Assuming there is a total of n cells, we define B_j as the batch label of cell j and b_j as the number of cells which has the same batch label as cell j . In the similarity matrix $\mathbf{Z} \in \mathbb{R}^{n \times n}$, given a certain cell j , the mean similarity between cells within the same batch equals $(\sum_{B_i=B_j}^n \mathbf{Z}_{ij})/b_j$ and the mean similarity between cells within different batches equals to $(\sum_{B_i \neq B_j}^n \mathbf{Z}_{ij})/(n - b_j)$. Typically, for a given certain cell, the mean similarity between cells within the same batch is significantly higher than that between different batches. To utilize cross-batch information for enhancement more effectively, we have introduced a fixed similarity matrix $\mathbf{Z}^{\text{fix}} \in \mathbb{R}^{n \times n}$.

$$\mathbf{z}_{ij}^{\text{fix}} = \begin{cases} \mathbf{z}_{ij} \times \frac{\frac{\left(\sum_{B_k \neq B_j}^n \mathbf{z}_{kj}\right)}{(n - b_j)}}{\frac{\left(\sum_{B_k = B_j}^n \mathbf{z}_{kj}\right)}{b_j} + \frac{\left(\sum_{B_k \neq B_j}^n \mathbf{z}_{kj}\right)}{(n - b_j)}}, & \text{if } B_i = B_j, \\ \mathbf{z}_{ij} \times \frac{\frac{\left(\sum_{B_k = B_j}^n \mathbf{z}_{kj}\right)}{b_j}}{\frac{\left(\sum_{B_k = B_j}^n \mathbf{z}_{kj}\right)}{b_j} + \frac{\left(\sum_{B_k \neq B_j}^n \mathbf{z}_{kj}\right)}{(n - b_j)}}, & \text{if } B_i \neq B_j. \end{cases}$$

In the extended scCASE, after the iterations of similarity matrix \mathbf{Z} , we weight it using the above equation, replace \mathbf{Z} with \mathbf{Z}^{fix} , ensuring that the mean similarity of each cell with cells from the same batch equals that with cells from different batches and achieved a better enhancement of data with batch effects. We provide the extended version of scCASE as an optional variant.

Following the reviewer's suggestion, we evaluated raw data and the data enhanced by various methods via two metrics of k-nearest neighbor batch effect test (kBET) and modified average silhouette width of batch (Batch ASW). The two metrics are provided by scib¹⁶ and reflect batch mixing degrees. kBET is a wrapper function of the implementation by Büttner et al²⁶, which measures the bias of a batch variable in the kNN graph, and Batch ASW measures the silhouette score of given batch labels¹⁶. kBET and Batch ASW are scaled by default between 0 and 1, in which larger scores represent better batch removal.

Taking the Mixed-protocols dataset as an example, this dataset exhibits obvious batch effects, with significant differences observed among L2/3 IT, L4, and L6 CT cells from different batches (Fig. R17b). Although the initial version of scCASE may not eliminate these batch effects, it still outperformed other methods in achieving improved clustering results in datasets with strong batch effects. Compared to other baseline methods, scCASE achieved the highest ARI, AMI, FMI, and Silhouette score (Fig. R17a). After extending scCASE using the aforementioned approach, there is a notable increase in similarity among L2/3 IT, L4, and L6 CT cells from different batches. This improvement is evident in cell clustering. Taking the L2/3 IT cells as an example, in the raw data and data enhanced by baseline methods, L2/3 IT cells from different batches exhibit significant differences, making it challenging to identify them as the same cell type in UMAP visualization and Louvain clustering (Fig. R17b-e). The extended scCASE is able to reduce the differences between L2/3 IT cells from different batches, resulting in their clustering as a single group in Louvain clustering (Fig. R17f). The extended scCASE achieved a significant lead in metrics measuring both the preservation of biological variation (ARI, AMI, FMI, and Silhouette score) and batch mixing (kBET and Batch ASW)

Fig. R17. Performance of scCASE and baseline methods in both preservation of biological variation and batch mixing on the Mixed-protocols dataset. **a**, The values of different metrics of the raw data and the data enhanced by various methods in the Mixed-protocols dataset. **b-f**, The UMAP visualization of the raw data and the data enhanced by various methods of the Mixed-protocols dataset.

(Fig. R17a). The results indicate that the extended scCASE can effectively correct batch effects and facilitate the analyses of data with batch effects.

In conclusion, on the one hand, we apologize for any ambiguous expressions in the manuscript. Our intent is not to assert that scCASE can address batch effects but rather to clarify that scCASE can effectively handle data in various scenarios. On the other hand, inspired by the reviewers, we explored the potential of scCASE in addressing batch effects. We have extended scCASE as an optional variant, enabling it to significantly enhance the analyses of data with batch effects. In the revised manuscript, we have modified Section “scCASE exhibits superior robustness to various application scenarios” and Supplementary Text S5. We once again express our gratitude for the valuable comment from the reviewer, which greatly contributed to the refinement of our manuscript.

COMMENT 8

Line 170: how are the cell embeddings generated? Implementation details and hyperparameters for each method?

RESPONSE TO COMMENT 8

Thank you for the valuable comment that contributes to refining the details in our manuscript. The implementation details for obtaining the cell embeddings at Line 170 (Fig. 3) are as follows. We first preprocessed the raw data and the data enhanced by different methods using TF-IDF. Then, we performed PCA to reduce the dimensionality, created a “neighbors” graph, and used t-SNE/UMAP to obtain two-dimensional visualizations of the data, following the scATAC-seq data analysis workflow provided by EpiScanpy²⁷ (https://colomemaria.github.io/episcanpy_doc/examples.html). The above steps were performed using the default parameters in the EpiScanpy pipeline. We have added the description in Section “Implementation details of downstream analyses”.

COMMENT 9

Fig. 4a: Are the column / row orders by hierarchical clustering? Or manually ordered. Are all peaks input to scCASE, or just differential peaks? If only differential peaks are used as inputs, isn't the analysis circular?

RESPONSE TO COMMENT 9

We apologize for any ambiguities in the manuscript. Figure 4a depicts a heatmap of the cell embedding matrix (H) obtained by scCASE. The columns are ordered based on cell labels. During the execution of the model, all peaks serve as inputs to the model. We again apologize for the ambiguities description and we have added this clarification in Section “scCASE intuitively reveals cell type-specific biological insights”.

COMMENT 10

Line 240: there is no description of how SNPsea analysis is performed and how to interpret the results. Where does the phenotype data come from? Are SNPs called on the scATAC dataset?

RESPONSE TO COMMENT 10

We apologize for not explaining the details of SNPsea analysis enough. In this revision, we provide detailed introduction and implementation details of SNPsea²⁸. SNPsea is an algorithm to identify cell types and pathways likely to be affected by risk loci. Specifically, genome-wide association studies (GWAS) have discovered multiple genomic loci associated with risk for different types of disease. SNPsea provides a simple way to determine the types of cells influenced by genes in these risk loci. SNPsea supposes disease-associated alleles influence a small number of pathogenic cell types, and assumes that a gene's specificity to a cell type is a reasonable indicator of its importance to the unique function of that cell type.

We performed SNPsea enrichment analysis with default settings in each set of cell type-specific peaks and the set of background peaks, respectively. The enrichments of tissue-specific expression in profiles of 17,581 genes across 79 human tissues (Gene Atlas) were quantified²⁹. Specifically, we first obtained SNP site data for the whole genome from HapMap3 SNPs, which can be downloaded at <https://zenodo.org/records/7768714>. To obtain the SNP sites corresponding to each group of cell type-specific peaks, we utilized GenomicRanges to identify SNP sites present within the cell type-specific peak regions³⁰. GenomicRanges is an R/Bioconductor package for representing and manipulating genomic intervals, available at <https://github.com/Bioconductor/GenomicRanges>. Then we obtained the SNP sites corresponding to each group of cell type-specific peaks, which served as the input for SNPsea. We specified the same additional data including phenotype data and parameters for SNPsea as in its tutorial. These additional data can be downloaded from <http://www.broadinstitute.org/mpg/snpsea> (SNPsea_data_20140520.zip). Their sources and detailed explanations are described at <https://snpsea.readthedocs.io/en/latest/data.html>. The parameters and specific running tutorials of the method can be found at <https://snpsea.readthedocs.io/en/latest/usage.html>. We quantified the enrichments of each set

of peaks in tissue-specific accessibility profiles across 79 tissues, and the top 30 significantly enriched tissues are illustrated in the Figures.

For readers to implement the same analysis as in the manuscript, we provide the specific code for implementing SNPsea in Zenodo¹¹, and provide the cell type-specific peaks (bed file) and corresponding SNP site data (anno file) as the example. We have added the above description in Section “Implementation details of downstream analyses”.

COMMENT 11

Line 249: LDSC implementation details not described. What data and phenotypes are used as input?

RESPONSE TO COMMENT 11

We apologize for the lack of implementation details of LDSC analysis. LDSC is a command line tool for estimating heritability and genetic correlation from GWAS summary statistics³¹. After identifying cell type-specific peaks and background peaks in the Blood dataset, we quantified the enrichment of heritability for blood-related phenotypes within cell type-specific peaks for each cell type using partitioned LDSC with default settings. We ran LDSC using HapMap3 SNPs and used European samples from the 1000 Genomes Project as the LD reference panel. All the summary statistics provided by LDSC, including SNPs and phenotypes, were downloaded from the Broad LD Hub (<https://doi.org/10.5281/zenodo.7768714>). Specifically, in this analysis, our input consists of the detected cell type-specific peaks or background peaks. Similar to SNPsea, we first utilized GenomicRanges to identify SNP sites present within the specific peak regions. Subsequently, we used the LDSC program to calculate the LD Scores for these SNP sites. The LDSC process in this step can be referred to at <https://github.com/bulik/ldsc/wiki/LD-Score-Estimation-Tutorial>. Finally, we invoked the LDSC program again, using the obtained LD Scores as input, to calculate heritability and genetic correlation with blood-related phenotypes. The LDSC process in this step can be referred to <https://github.com/bulik/ldsc/wiki/Heritability-and-Genetic-Correlation>.

To facilitate readers in replicating the same analysis as the manuscript, we have provided the code for LDSC implementation on Zenodo¹¹. Additionally, examples of peak files (bed file), corresponding SNP site data (anno file), and LD score files (ldscore.gz file) are included. We have added the above description in Section “Implementation details of downstream analyses”.

COMMENT 12

Line 261: “we identified 1000 specific peaks ... based on the enhanced matrix”. Why does the authors choose to do clustering based using enhanced matrix + scABC method? How about the cell embeddings learn by scCASE? Are the learnt cell embeddings useful for clustering?

RESPONSE TO COMMENT 12

We sincerely apologize for any confusion caused by the confusing description in the first version of our manuscript. As clarified in Response to Comment #5 of Reviewer #1, in our manuscript, we solely performed PCA on the enhanced data, followed by Louvain clustering to obtain the cluster labels. We did not choose to do clustering based on the enhanced matrix + scABC. The cell embeddings learned from the scCASE model were utilized for identifying cell-type specific peaks in downstream analyses.

Fig. R18. Evaluation of clustering performance using cell embeddings of scCASE and other scCAS data enhancement methods. **a**, The clustering performance evaluated by ARI. **b**, The clustering performance evaluated by AMI. The term "scCASE enhancement" refers to clustering using scCASE enhanced data + PCA.

The insightful comment from the reviewer regarding the utility of learned cell embeddings for clustering has inspired our subsequent analysis. Specifically, we applied Louvain clustering to the learned cell embeddings and compared the results with clustering based on PCA of the enhanced data. The findings indicate that the clustering result of cell embedding is comparable to the clustering results obtained through the enhanced data + PCA (Fig. R18). Moreover, cell embeddings learned by scCASE exhibit certain advantages over the latent representations obtained from other data enhancement methods. Compared to the latent representations obtained from baseline methods, the learned cell embeddings by scCASE provided the overall best clustering performance. However, it is crucial to note that our method is designed for enhancement rather than dimensionality reduction. Therefore, our primary focus lies in evaluating the clustering results of the enhanced data. We have added the above results in Section “scCASE enhances scCAS data for better cellular heterogeneity characterization”.

COMMENT 13

Line 263: scABC implementation not described.

RESPONSE TO COMMENT 13

We apologize for not explaining the details of scABC analysis enough. Following the approaches of RA3¹², we used scABC to identify cluster-specific peaks. scABC is an R package for the analysis of scATAC-seq data³². With the clustering assignments obtained from the data enhanced by scCASE and Louvain clustering, we followed the scABC workflow, utilized the function “getClusterSpecificPvalue()”, calculated the *p*-value using hypothesis testing procedure, and finally identified cluster-specific peaks (<https://github.com/SUwonglab/scABC/blob/master/vignettes/ExampleWorkflow.html>). These identified cluster-specific peaks will be used as the input of chromVAR³³ to perform motif analysis similar to RA3¹². We have added the above description in Section “Implementation details of downstream analyses”.

COMMENT 14

Line 263: chromVAR implementation not described. What motif base is used?

RESPONSE TO COMMENT 14

chromVAR is an R package for the analysis of sparse chromatin accessibility data from single-cell/bulk ATAC-seq/DNase-seq³³. The package aims to identify motifs or other genomic

annotations associated with variability in chromatin accessibility between individual cells or samples. We downloaded chromVAR from <https://greenleaflab.github.io/chromVAR/>.

The motifs database is obtained from the “getJasparMotifs()” function within the chromVAR method, which is sourced from the JASPAR³⁴ database. Following the workflow in RA3¹², we used the data enhanced by scCASE as the input and applied chromVAR to infer the enriched transcription factor (TF) binding motifs within the top 1000 cluster-specific peaks with the smallest *p*-values calculated by scABC. Subsequently, we visualized the deviations calculated by chromVAR for the top 50 TF binding motifs. We have added the above description in Section “Implementation details of downstream analyses”.

COMMENT 15

Line 304: “We randomly subsampled ...”. Consider also scale up the dataset size to test memory and runtime limitation of the model on large datasets.

RESPONSE TO COMMENT 15

We appreciate the reviewer for this suggestion. To validate the scalability of our method on larger datasets, we conducted run-time and peak memory usage analyses on datasets of varying sizes. We compared the run-time and peak memory usage of scCASE with that of baseline methods and demonstrated the method's scalability on larger datasets. We provided the details in Response to Comment #3 of Reviewer #1.

REVIEWER #2

Tang et al present scCASE, a new computational method for modeling single-cell chromatin accessibility data that enhances data analyses via non-negative matrix factorization. Through a range of benchmarks on real and simulated data, the authors argue that scCASE has superior performance.

RESPONSE

We express our sincere gratitude to the reviewer for the constructive comments which helped us to improve the manuscript. We have carefully considered and answered every comment raised by the reviewer and made revisions to the manuscript correspondingly.

COMMENT 1

There are now ~dozens of methods that perform similar analyses. The authors provide benchmarking that shows overall metrics like ARI but provide very limited biological intuition for why their methods outperforms any state of the art workflows.

RESPONSE TO COMMENT 1

We appreciate the reviewer for this thoughtful comment. Firstly, we agree with the reviewer's statement that "There are now ~dozens of methods that perform similar analyses." There are numerous enhancement methods for scRNA-seq data. Nevertheless, only a limited number claim to be suitable for scCAS data³⁵. Due to the substantial differences between scRNA-seq and scCAS data, particularly the increased sparsity and higher dimensionality of scCAS data, there are currently few methods that can effectively enhance scCAS data. Applying scRNA-seq enhancement methods directly to scCAS data results in suboptimal and non-robust performance^{10, 36}. The existing scCAS data enhancement methods overlook the crucial factors of cell-to-cell difference and correlation and are unable to harness the large compendia of available omics data, and these limitations constrain their performance. Therefore, there is still a lack of scCAS data enhancement methods specifically designed for enhancing scCAS data.

Secondly, as a method designed for biological research, we deeply agree with the reviewer that having a biological interpretation behind the model is crucial. scCASE is based on non-negative matrix factorization (NMF). NMF decomposes data into latent components and has clear interpretability as basis vectors for the latent factors. For example, in the scCASE model, the projection matrix (**W**) stores weights regarding the patterns of peaks, while the cell embedding matrix (**H**) holds weights for the patterns of cells. In this context, the chromatin accessibility information for each cell is represented as a weighted summation of multiple components. Thus, scCASE can extract latent features of cell subpopulations, which can be applied to downstream analyses and reveal cell type-specific biological insights (Fig. 4a).

Moreover, allowing negative values in the models may lead the algorithm to subtract real data instead of imputing the missing ones. A specific characteristic of NMF is that the matrices obtained through its decomposition do not allow for the presence of negative values. Therefore, NMF can achieve better imputing in scenarios where the matrix to be factorized contains generous dropout events such as scCAS data.

Thirdly, compared to existing methods, scCASE demonstrates innovation in two key aspects, providing it with distinct advantages. On one hand, scCASE explicitly leverages the similarity between cells. Given that similar cells generally have similar chromatin accessibility patterns, in comparison to other methods, scCASE introduces a cell-to-cell similarity matrix, allowing the chromatin accessibility of one cell to be represented as a weighted combination of that of other cells. On the other hand, existing scCAS enhancement methods neglect the wealth of existing data in public databases. scCASE is capable of incorporating publicly available omics data as reference data, providing prior knowledge to better characterize the target scCAS data and facilitate data enhancement.

Finally, scCASE can extract latent features of cell subpopulations, which can be applied to downstream analyses and reveal cell type-specific biological insights. Specifically, scCASE can identify cell type-specific peaks, allowing us to gain insights into various aspects of cellular heterogeneity, such as functionality, tissue-specific expression, and partitioned heritability. We performed downstream analyses, including SNPsea enrichment analysis²⁸ (Fig. 4b), LDSC heritability enrichment analysis³¹ (Fig. 4c), GREAT genomic region enrichment analysis³⁷, and motif analysis³³ (Fig. 4d). The results illustrated that scCASE can reveal cell type-specific biological insights, learn biological features from data, and outperform other workflows.

In conclusion, we have discussed the biological interpretations of the model's effectiveness both from the perspective of method design and downstream analysis. Once again, we express our gratitude for the valuable comment from the reviewer. We have added the above discussion to Supplementary Text S7.

COMMENT 2

If this NMF workflow has a meaningful improvement in learning biological signal over a more typical PCA / LSI implementation, can the authors unpack this?

RESPONSE TO COMMENT 2

We appreciate the reviewer for this insightful comment. We will elucidate the improvement of scCASE in learning biological signals over typical PCA / LSI through the following aspects: 1) the advantages of NMF over PCA and LSI in uncovering biological significance, 2) the distinctions between data enhancement methods such as scCASE and data dimensionality

reduction methods such as PCA and SVD, 3) the biological interpretation of the innovative aspects of scCASE base on NMF.

Fig. R19. Clustering performance of the data enhanced by PCA, LSI and scCASE.

Principal component analysis (PCA) is a widely used dimensionality reduction method. The primary goal of PCA is to transform the original high-dimensional dataset into a new coordinate system, where the data's variance is maximized along the principal components. These components are orthogonal to each other, and the top few components capture the most significant variance in the data. Latent semantic indexing (LSI) is a technique primarily used in natural language processing and information retrieval. By using singular value decomposition (SVD), LSI identifies latent semantic structures in the data, allowing for more effective document retrieval and information organization. Both PCA and LSI are effective for scCAS data dimensionality reduction, but they are less commonly applied to data enhancement tasks. For example, the pipeline in EpiScanpy provides steps that utilize PCA for dimensionality reduction rather than for imputation. We performed imputation on various datasets using either PCA or LSI, and evaluated the imputed data by different clustering metrics. The results indicate that compared with scCASE, PCA and LSI fall short of achieving effective data enhancement (Fig. R19). NMF produces factor matrices that are non-negative and have clear interpretability as basis vectors for the latent factors. As mentioned in Response to Comment#1 of Reviewer #2, NMF decomposes data into latent components and has clear interpretability as basis vectors for the latent factors. For example, in the scCASE model, the projection matrix (**W**) stores weights regarding the peak-component, while the cell embedding matrix (**H**) holds weights for the cell-component. In this context, the chromatin accessibility information for each cell is represented as a weighted combination of multiple components. Moreover, NMF breaks down a non-negative matrix into the product of lower-rank non-negative matrices, approximating the original matrix while preserving this non-negativity. It means that NMF optimizes the reconstruction loss, similar to autoencoders, which is distinct from PCA and LSI. Non-negativity constraint also promotes the imputation of missing signals,

as signals are explained additively in one direction. If negative values are permitted in the models, the algorithm may be tempted to subtract away real data points rather than imputing missing points. The abovementioned features make NMF commonly employed for imputation, and NMF has demonstrated its efficacy in data enhancement³⁸⁻⁴¹.

Secondly, we would like to underscore that scCASE is an enhancement method primarily designed to impute and denoise the original scCAS data, focusing on preserving the data's original dimensions rather than achieving dimensionality reduction. In contrast, PCA and LSI are employed for data dimensionality reduction, providing a low-dimensional representation of cells that can be utilized for downstream analyses such as clustering. These two types of methods are not mutually exclusive but can complement each other. For instance, scCASE acts as a preprocessing step to enhance the original data in its original dimensions, while PCA can be subsequently applied to the enhanced data for dimensionality reduction, obtaining a cell embedding. In our manuscript, following the approach of scOpen, we utilized data enhancement methods to impute the original data and then applied PCA to the enhanced data to obtain the cell embedding, followed by clustering and evaluation of the cell embedding.

Finally, scCASE is not a simple NMF model. In the modeling process, scCASE additionally considers the similarity between cells and creates a cell similarity matrix to smooth the data using cell similarity. This means that the chromatin accessibility state of a cell can be represented as the weighted average of the accessibility states of similar cells. Through multiple iterations, the projection matrix, cell embedding matrix, and similarity matrix are continuously updated, combining this information to achieve data enhancement. Moreover, scCASE is capable of incorporating publicly available omics data as reference data, providing prior knowledge to better characterize the target scCAS data and facilitate data enhancement.

We express our gratitude for this thoughtful comment. We have added the abovementioned discussion to Supplementary Text S7.

COMMENT 3

Most of the benchmarking is done on datasets with limited cell numbers (e.g. profiling via the fluidigm c1) but contemporary datasets scale to 100,000+ cells. More benchmarking / analyses on large datasets profiled via 10x scATAC-seq is required to understand scCASE's utility with modern datasets.

 related to this, what are the considerations for scCASE's computational efficiency as datasets get larger?

RESPONSE TO COMMENT 3

Thank you for this constructive comment, which highlighted the importance of computational efficiency. We deeply appreciate your perspective on the significance of run-time and memory efficiency for a computational method.

Firstly, we evaluated the computational efficiency of the original implementation of scCASE. The run-time and peak memory usage on different datasets using the original scCASE are provided in Fig. R20a. As a method that explicitly considers cell-to-cell similarity, as mentioned by Reviewer #1, for larger datasets, the number of cells can be a bottleneck for the computational efficiency of scCASE. This is because scCASE calculates the similarity matrix between cells and uses it for matrix operations during iterations. We acknowledge that the original version of scCASE did not have a computational efficiency advantage over other methods. Therefore, in this revision, we have made optimizations to the scCASE code. 1) We changed the calculation method of Jaccard similarity, using sparse matrix operations instead of the Jaccard calculation method provided by Scipy. 2) We adjusted the precision of data storage

Fig. R20. Run-time and peak memory usage. **a**, Run-time and peak memory usage of the original implementation of scCASE and the updated implementation of scCASE. **b**, Run-time and peak memory usage of updated scCASE and baseline methods.

from double-precision floating-point to single-precision floating-point. This approach reduced memory usage and did not impact experimental results. 3) We refactored the matrix computation process in the code, employing faster matrix operation functions and reducing the use of intermediate variables to expedite the computation of iteration results and reduce memory consumption. These optimizations solely reduce run-time and memory usage and will not have any impact on the results presented in the manuscript. The comparison of run-time and peak memory usage of scCASE before and after the optimizations on different datasets are illustrated in Fig. R20a, b. These optimizations significantly reduced peak memory usage and run-time consumption. On the Blood and LungA datasets, the average reduction in peak memory usage is about 30%, and the run-time is reduced by approximately 80%, our optimizations have a more pronounced effect on larger datasets. For example, on the larger Muto dataset, the updated scCASE requires only half the original memory and one-tenth of the original time.

Next, we tested the updated scCASE on varying datasets, extending the maximum number of cells to 100k. We compared the updated scCASE with other baseline methods in terms of run-time and memory usage, the optimized scCASE demonstrates satisfactory performance in both run-time and memory usage. Specifically, the hardware configuration utilized in our experiments is outlined below: 500GB memory, Intel Xeon Gold 6348 CPU @ 2.60GHz with 112 cores, Ubuntu 22.04.2 LTS. For SCALE and scBasset, two deep learning-based methods, we used NVIDIA A40 with 48GB memory. In terms of run-time, scCASE exhibits significant advantages compared to other methods, especially on smaller datasets such as BM0828, Blood, and LungA, where scCASE can operate several times faster than baseline methods (Fig. R20c). Even on larger datasets, scCASE still maintains a notable speed advantage (Fig. R20c). In terms of peak memory usage, SCALE and scBasset, two GPU-based methods, make more usage of GPU memory, and thus typically require less memory usage than the methods that utilize CPUs. scCASE demonstrates a certain advantage in peak memory usage on smaller datasets (BM0828, Blood, and LungA) (Fig. R20d). Although the peak memory usage of scCASE increases in larger datasets, its memory usage is still comparable to that of scOpen, the state-of-the-art scCAS data enhancement method (Fig. R20d). Moreover, the memory usage of scCASE growth remains manageable. On two larger datasets of Muto and Simulated, which have a similar number of peaks but a fivefold increase in the number of cells (from 20k to 100k), the peak memory usage of scCASE increased by 7.67 times, while that of scOpen increased by 8.62 times. Note that scBFA is unable to run on datasets with 100k cells due to out-of-memory errors.

In conclusion, we have improved the computational efficiency of scCASE and conducted a comparative benchmark of its run-time and peak memory usage against baseline methods.

The improved scCASE demonstrates notable scalability in terms of both run-time and memory usage compared to other methods. In the revised manuscript, we have included the results in Section “Run-time and memory usage of scCASE” (Supplementary Fig. S29).

COMMENT 4

If I understand correctly, a 1% threshold for the counts matrix has to be applied to run scCASE-- this seems problematic as it would discard thousands of peaks. What's the impact of this? Is this required for computational efficiency? A more careful analyses of the upstream QC is required.

RESPONSE TO COMMENT 4

Thank you for the meticulous review and for this insightful comment. In scCASE, the filtering of peaks and the threshold for filtering are treated as optional parameters. In other words, this is not a mandatory step. However, in the analysis of scCAS data, the standard workflow of EpiScanpy and the methods specifically designed for scCAS data usually include peak filtering during the preprocessing steps^{8, 9, 12, 14, 21, 36}. This is because scCAS data often contains numerous peaks only accessible in very few cells, or even completely inaccessible in all cells. To validate the impact of peak filtering and different filtering thresholds on clustering outcomes, we conducted the following experiments.

Firstly, we executed the scCASE method without peak filtering on eight datasets and evaluated both computational efficiency and clustering performance. The experimental results demonstrated that the clustering results using the unfiltered scCASE method and the scCASE method filtered with a default 1% threshold were relatively close (Fig. R21a, b). We performed a two-sided Wilcoxon signed-rank test on the clustering metrics of scCASE with a 1% filtering threshold and scCASE without filtering. The *p*-values for ARI were 0.84, and for AMI it was 0.64, which indicated that the 1% filtering threshold does not significantly impact the clustering metric. Simultaneously, the removal of these peaks effectively accelerated the run-time and saved memory usage. Across the BM0828, Blood, and LungA datasets, a 1% filtering threshold results in an average reduction of 56% (Fig. R21c) in run-time and 69% in peak memory usage (Fig. R21d).

Secondly, we also explored the impact of different filtering thresholds on scCASE. We ran scCASE with filtering thresholds set at 3% and 5% on the eight datasets, assessing both computational efficiency and clustering performance. The results indicated that compared to the 1% filtering threshold, using 3% and 5% thresholds significantly discarded valuable information in the datasets, leading to a decrease in clustering metrics (Fig. R21a, b). We acknowledge that filtering can indeed lead to information loss. However, it is crucial to

emphasize that this is not a case of "the more, the better" or "the less, the better." All the experiments in our manuscript indicate that the default 1% filtering threshold is a reasonably suitable choice.

Fig. R21. The analyses of different peak filtering strategies. a, The ARI of clustering with varying peak filter threshold. **b,** The AMI of clustering with varying peak filter threshold. **c,** The run-time with varying peak filter threshold. **d,** The peak memory usage with varying peak filter threshold.

Finally, we assessed the impact of the filtering on the identification of cell type-specific peaks. We ran scCASE without peak filtering on the Blood dataset and utilized scCASE to identify 1000 specific peaks for each cell type. Next, we examined whether those cell type-

specific peaks were discarded when using a 1% filtering threshold. In most cases, the 1000 cell type-specific peaks obtained by the unfiltered scCASE method were not discarded during the filtering process. 99.56% of peaks were saved on average. This indicates that the filtering strategy does not significantly impact the identification of cell type-specific peaks.

In conclusion, firstly, the threshold for filtering is treated as an optional parameter in scCASE, allowing users to choose whether to perform filtering based on their specific needs. Secondly, filtering for the scCAS count matrix is a widely employed strategy, and our experiments demonstrate that a default 1% filtering threshold can decrease run-time and memory usage without affecting cell type identification and downstream analyses. Once again, we appreciate the comment provided by the reviewer, and we have included the results in Supplementary Text S6.

COMMENT 5

The GitHub provides minimal working examples of the code but lags behind the more interactive vignettes of other tools (e.g. SCALE, Seurat/Signac, etc.). For adoption of the workflow, more detailed step-by-step instructions explaining the functions called is required. Further, cleaner integration with tools like scanpy and Signac would allow for much greater adoption.

RESPONSE TO COMMENT 5

We appreciate the reviewer's valuable comment. As a paper primarily focused on computational methods, we apologize for not providing sufficient tutorials and workflows. In response to the reviewer's feedback, we have modified the code of scCASE to make it more user-friendly. Moreover, we have modularized the overall functionality of scCASE, introducing additional functions to facilitate its step-by-step application in data processing workflows. Furthermore, we have diligently revised the tutorial, providing a detailed, step-by-step description of each invoked function along with its functionalities and significance which aims to offer users a more thorough understanding of the function and usage of scCASE. Furthermore, we have included an additional tutorial to facilitate the integration of scCASE with EpiScanpy²⁷, the widely used pipeline for single-cell epigenomic data.

REVIEWER #3

Single-cell chromatin accessibility sequencing provides rich data for analyzing epigenetic information and gene regulation mechanisms at the single-cell level. However, due to the sparsity and dropout events of single-cell sequencing data, how to accurately and efficiently obtain accessibility peak information has become a top priority in analyzing and applying single-cell chromatin accessibility sequencing data. In this manuscript, the authors applied non-negative matrix factorization to enhance single-cell chromatin accessibility sequencing (scCAS) data, proposing scCASE and scCASER that integrate external reference data. Compared to many published methods, it significantly enhances scCAS data, improves the credibility and interpretability of accessibility peaks, and maintains good robustness when facing different data scales and batches. This will facilitate the effective use of scCAS data and enhance its interpretability.

However, several concerns require the author's response and explanation when revising the manuscript:

RESPONSE

We would like to express our gratitude to the reviewer for the meticulous review and valuable comments. These comments have played a crucial role in enhancing our manuscript. We have carefully considered and addressed each comment, and have made revisions to the manuscript in the corresponding sections.

COMMENT 1

1. scCASE and scCASER are both constructed based on the NMF method. What are the advantages of the number of CPU cores, GPU cores, and processing time required to enhance scCAS data of the same scale compared to other algorithms used as comparative objects in other manuscripts? I did not see this part of the data in the manuscript.

RESPONSE TO COMMENT 1

Thank you for this insightful comment. The reviewers' particular attention to computational efficiency has underscored the importance of this aspect in evaluating a computational method. Given the reviewers' heightened concern for computational efficiency, in this revision, we have updated and optimized the scCASE code in its implementation process. First, we changed the method of calculating Jaccard similarity, replacing the original Jaccard calculation provided by Scipy with sparse matrix operations. Second, we adjusted the precision of data storage from double-precision floating-point to single-precision floating-point, which reduced memory usage and did not impact experimental results. Third, we refactored the matrix computation

process in the code, employing faster matrix operation functions and reducing the use of intermediate variables to expedite the computation of iteration results and reduce memory consumption. These optimizations solely reduce run-time and memory usage and will not have any impact on the results presented in the manuscript. The efficiency of the improved scCASE has significantly increased compared to the original version. Specifically, on the Blood and LungA datasets, there is an average reduction of approximately 80% in run-time (Fig. R22a) and an average reduction of approximately 30% in peak memory usage (Fig. R22b). Meanwhile, on the larger Muto dataset, the improved scCASE required only half the original peak memory usage and one-tenth of the original run-time (Fig. R22a, b).

Additionally, following reviewers' comments, we conducted extensive testing on diverse datasets, expanding the maximum number of cells to 100k. The hardware configuration utilized in our experiments is outlined below: 500GB memory, Intel Xeon Gold 6348 CPU @ 2.60GHz with 112 cores, Ubuntu 22.04.2 LTS. For SCALE and scBasset, two deep learning-based methods, we used NVIDIA A40 with 48GB memory. In our evaluations, we compared the optimized scCASE with other baseline methods in terms of average CPU core usage, run-time, and peak memory usage. scCASE, as a method primarily utilizing matrix operations for

Fig. R22. Run-time, peak memory usage and average CPU core usage of scCASE and baseline methods. a, Run-time and **b**, peak memory usage of the original implementation of scCASE and the updated implementation of scCASE. **c**, Average CPU cores usage, **d**, run-time, and **e**, peak memory usage of updated scCASE and baseline methods.

iteration, only employs the CPU during computation and can utilize multiple cores for parallel acceleration, thus exhibiting a high CPU core usage (Fig. R22c). In terms of run-time, scCASE exhibits significant advantages compared to other methods, especially on smaller datasets such as BM0828, Blood, and LungA, where scCASE can operate several times faster than baseline methods (Fig. R22d). Even on larger datasets, scCASE still maintains a notable speed advantage (Fig. R22d). In terms of peak memory usage, SCALE and scBasset, two GPU-based methods, make more usage of GPU memory, and thus typically require less memory usage than the methods that utilize CPUs. scCASE demonstrates a certain advantage in peak memory usage on smaller datasets such as BM0828, Blood, and LungA (Fig. R22e). Although the peak memory usage of scCASE increases in larger datasets, its memory usage is still comparable to that of scOpen (Fig. R22e). Moreover, the memory usage of scCASE growth remains manageable. On two larger datasets of Muto and Simulated, which have a similar number of peaks but a fivefold increase in the number of cells (from 20k to 100k), the peak memory usage of scCASE increased by 7.67 times, while that of scOpen increased by 8.62 times. Note that scBFA is unable to run on datasets with 100k cells due to out-of-memory errors (Fig. R22e).

In conclusion, we have improved the computational efficiency of scCASE and conducted a comparison of its average CPU core usage, run-time, and peak memory usage with that of the baseline methods. The improved scCASE exhibits a distinct advantage in terms of run-time and memory usage when compared to other methods, and has the capability to leverage multiple CPU cores for parallel computation, thereby reducing the run-time. The detailed results are presented in Section “Run-time and memory usage of scCASE” in the revised manuscript (Supplementary Fig. S29).

COMMENT 2

2. PCA and SVD are also commonly used high-dimensional data dimensionality reduction methods. What are the advantages of enhancing NMF in scCAS data compared to these two methods?

RESPONSE TO COMMENT 2

We appreciate the reviewer for this thoughtful comment. We agree with the reviewer's perspective that PCA and SVD are commonly used methods for data dimensionality reduction. In the following sections, we will elaborate on the advantages of scCASE in scCAS data enhancement from three key aspects: 1) the algorithmic principles, characteristics, and advantages of NMF compared to PCA and SVD, 2) the distinctions between data enhancement methods such as scCASE and data dimensionality reduction methods such as PCA and SVD, and 3) the innovative aspects of scCASE building upon NMF.

Firstly, the PCA process involves finding a set of standard orthogonal bases, where the first base represents the direction with the highest variance in the original dataset. Subsequently, each subsequent base is chosen to be orthogonal to the previous ones and captures the maximum variance in its direction. Therefore, each dimension in PCA only represents the magnitude of variance, lacking clear physical interpretation for each base. When applied to scCAS data, PCA can reduce the dimensionality of the data but fails to extract cell type-specific chromatin accessibility patterns. The elements in the matrix obtained through PCA can be positive or negative, and since chromatin accessibility states are binary (open or closed) and the negative values cannot effectively describe this, making it challenging to interpret the results of PCA. Similarly, PCA can be seen as a specific application of SVD, sharing similar characteristics. The principal components obtained through PCA are essentially the left singular vectors of the data matrix. Both PCA and SVD are effective for data dimensionality reduction, but they are less commonly applied to data enhancement tasks. The standard workflow in EpiScanpy also provides steps using PCA for dimensionality reduction rather than enhancement. We performed imputation on various datasets using either PCA or SVD following the evaluation workflow as scOpen¹⁰. We also assessed the clustering metrics. Figure R23 indicated that compared with scCASE, they also fall short of achieving effective data enhancement.

In contrast to PCA and SVD, NMF decomposes data into latent components and has clear interpretability as basis vectors for the latent factors. For example, in the scCASE model, the projection matrix (**W**) stores weights regarding the peak-component, while the cell embedding matrix (**H**) holds weights for the cell-component. In this context, the chromatin accessibility information for each cell is represented as a weighted combination of multiple components. Moreover, NMF breaks down a non-negative matrix into the product of lower-rank non-negative matrices, approximating the original matrix while preserving this non-negativity. It

Fig. R23. Clustering performance of the data enhanced by PCA, SVD and scCASE, respectively.

means that NMF optimizes the reconstruction loss, similar to autoencoders, which is distinct from PCA and LSI. Non-negativity constraint also promotes the imputation of missing signals, as signals are explained additively in one direction. If negative values are permitted in the models, the algorithm may be tempted to subtract away real data points rather than imputing missing points. These characteristics make NMF commonly utilized in imputation tasks³⁸⁻⁴¹.

Secondly, we would like to emphasize that scCASE is a scCAS data enhancement method designed primarily to address dropout events in scCAS data. Its purpose is to impute and denoise the original scCAS data, focusing on preserving the data's original dimensions rather than achieving dimensionality reduction. In contrast, PCA and SVD are employed for data dimensionality reduction, providing a low-dimensional representation of cells that can be utilized for downstream analyses such as cell clustering. These two types of methods are not mutually exclusive but can complement each other. For instance, scCASE acts as a preprocessing step to enhance the original data in its original dimensions, while PCA can be subsequently applied to the enhanced data for dimensionality reduction, obtaining a cell embedding. In our manuscript, following the approach of scOpen¹⁰, we utilized data enhancement methods to impute the original data and then applied PCA to the enhanced data to obtain the cell embedding, followed by clustering and evaluation of the cell embedding.

Finally, scCASE is not a simple NMF model. In the modeling process, scCASE additionally considers the similarity between cells and creates a cell similarity matrix to smooth the data using cell similarity. This means that the chromatin accessibility state of a cell can be represented as the weighted average of the accessibility states of similar cells. Through multiple iterations, the projection matrix, cell embedding matrix, and similarity matrix are continuously updated, combining this information to achieve data enhancement. Additionally, scCASE has the capability to integrate publicly available omics data as reference data. This integration offers prior knowledge to enhance the characterization of the target scCAS data and facilitate data enhancement.

We express our gratitude for the insightful comment. We have added the discussion of the advantages of scCASE compared to PCA and SVD to Supplementary Text S7.

COMMENT 3

3. Figure 4a and Supplementary Figure 7a in the manuscript are the same. Is it an mistake or why show it two times?

RESPONSE TO COMMENT 3

We appreciate the reviewer's meticulous comment. Figure 4a and Supplementary Fig. 7a are identical. During the manuscript composition, we deemed the content displayed in these two images to be relatively similar, both effectively illustrating the functionality of scCASE in identifying cell-type-specific peaks. Placing Fig. 4a within Supplementary Fig. S7 was done to avoid inappropriate scaling or whitespace on the respective page. Additionally, this arrangement facilitates the ease of reference to Fig. 4a when reading the supplementary Figures. Therefore, for the sake of aesthetics and completeness, we have included a duplicate of Fig. 4a in Supplementary Fig. S7. In the revision of the manuscript, Supplementary Fig. S7a has been removed.

COMMENT 4

4. The results shown in Figure 4a and Supplementary Figure 7 in the manuscript indicate that scCASE can effectively reveal biological insights on cell type specificity, but I have not seen corresponding results from other methods, which seem to fail to effectively demonstrate that scCASE has more advantages compared to other methods. How many overlapping peaks of cell type specificity can be obtained by different methods for the same data source?

RESPONSE TO COMMENT 4

We appreciate the valuable comments from the reviewer. Firstly, our description of downstream analyses aims to demonstrate that scCASE not only can effectively enhance scCAS data but also is an interpretable method. Its advantage lies in the ability to enhance data while simultaneously uncovering biological insights about cellular heterogeneity. For scCASE, we can consider a column of the projection matrix (a pattern of peaks) corresponding to the row of cell embedding with the highest activation levels in a certain cluster, investigate the pattern of peaks with relatively large coefficients, and identify the cell type-specific peaks. Specifically, the functionality of scCASE in identifying specific peaks is derived from the interpretability of its model itself, rather than being a result of combining with other tools, while many other methods can only enhance data and cannot reveal the biological insights.

Secondly, following the reviewer's comment, we comprehensively compare the cell type-specific peaks identified by scCASE and the differentially accessible peaks (DAPs) identified by EpiScanpy. Taking the Blood dataset as an example again, we computed the overlap between the scCASE-identified cell type-specific top 1000 peaks and the top 1000 DAPs identified by EpiScanpy. Figure R24 illustrates a moderate degree of overlap between the cell type-specific peaks identified by scCASE and the DAPs identified by EpiScanpy. In six cell types, including CLP, CMP, LMPP, MEP, mono, and pDC, 50%-80% of the peaks are

Fig. R24. The overlap of cell type-specific peaks identified by scCASE and the differential accessible peaks identified by EpiScanpy on the raw data.

overlapped, and there is less degree of overlap in the cell types of HSC and GMP, respectively with 341 and 343 overlapping peaks.

Due to the limited overlap between cell type-specific peaks identified by scCASE and the DAPs identified by EpiScanpy of HSC and GMP cells, we used these two types as an example to demonstrate how the specific peaks identified by scCASE contribute to superior biological insights into cellular heterogeneity. The two methods had four sets of peaks across the two cell types, including HSC-specific peaks identified by scCASE, GMP-specific peaks identified by scCASE, DAPs of HSC identified by EpiScanpy and DAPs of GMP identified by EpiScanpy. To investigate the differences in the peaks obtained by the two methods, we removed their intersection in each type, resulting in an additional four sets of peaks, including scCASE-unique HSC-specific peaks, scCASE-unique GMP-specific peaks, the unique DAPs of HSC identified by EpiScanpy, and the unique DAPs of GMP identified by EpiScanpy. Then we performed single-nucleotide polymorphisms (SNPs) enrichment analysis using SNPsea to obtain tissues explicitly affected by these peaks. Note that hematopoietic stem cells (HSCs) serve as the foundational source for immune cells, including T cells and B cells. GMP (granulocyte-macrophage progenitor) cells represent a stage in hematopoiesis and give rise to various immune cells. HSCs and GMPs play a central role in orchestrating the generation and continuous replenishment of various immune cell types, contributing to the overall functionality of the immune system. Therefore, the HSC and GMP specificity peaks should exhibit a higher correlation with whole blood, myeloid cells, and lymphocytes. In the specific peaks obtained by scCASE, we can significantly observe this correlation (Fig. R25a-d).

Fig. R25. SNPsea enrichment analysis for the cell type-specific peaks identified by scCASE and the differential accessible peaks identified by EpiScanpy on the raw data. **a-d**, scCASE-identified cell type-specific peaks. **e-h**, DAPs identified by EpiScanpy.

However, the specific peaks unearthed by EpiScanpy do not capture this correlation effectively (Fig. R25e-h). The genomic region enrichment of annotation tool (GREAT) analysis of scCASE-unique HSC-specific peaks obtained 20 pathways, comprehensively associated with functions such as immune regulation, immune cell activation, hematopoietic regulation, etc. In comparison, the unique DAPs of HSC identified by EpiScanpy consist of only four pathways, solely related to immune regulation.

Finally, we applied EpiScanpy to the data enhanced by baseline methods to identify DAPs. We first generated an upset plot for each group of peaks, and the results showed a higher intersection between EpiScanpy + raw data and scCASE, while the intersections between EpiScanpy + data enhanced by other methods were less prominent (Fig. R26). To validate whether EpiScanpy can identify DAPs with greater biological specificity from data enhanced by baseline methods, we utilized DAPs identified by EpiScanpy on the raw data, DAPs identified by EpiScanpy on the data enhanced by scOpen, and cell type-specific peaks identified by scCASE as examples, given the DAPs identified by EpiScanpy + scOpen had minimal overlap with that by EpiScanpy + raw data (Fig. R26). We generated heatmaps using the raw data and different sets of peaks (Fig. R27). Figure R27a displays DAPs identified by EpiScanpy on the raw data, Fig. R27b showcases DAPs identified by EpiScanpy in the data enhanced by scOpen, and Fig. R27c presents cell type-specific peaks identified by scCASE. It was observed in Fig. R27b that when EpiScanpy was applied to the data enhanced by scOpen, although the specific peaks obtained were indeed specific, compared to the original data, these peaks exhibited low accessibility in each type and higher randomness. Therefore, they are not the biological cell type-specific peaks we aimed to identify. In other words, the accessibility of the identified peaks is limited, and they are only sporadically accessible in certain cells, rather than being specific to that particular cell type, holding less biological significance. The enhancement by scOpen magnified such signals, which is not desired. Downstream analysis also confirmed that these peaks do not effectively reflect cellular heterogeneity. We implemented downstream analysis using cell type-specific peaks identified by scCASE and the DAPs identified by EpiScanpy + data enhanced by baseline methods. With regard to SNPs enrichment analysis, HSC-specific peaks and GMP-specific peaks identified by scCASE demonstrate better cell type-specificity compared to EpiScanpy + data enhanced by baseline methods (Fig. R28). Moreover, the HSC-specific peaks and GMP-specific peaks identified by scCASE showed higher associations with blood-related phenotypes than that of EpiScanpy + data enhanced by baseline methods (Fig. R29).

Fig. R26. The overlap of cell type-specific peaks identified by scCAGE and the differential accessible peaks identified by EpiScanpy on raw data and data enhanced by baseline methods.

Fig. R27. Cell-peak heatmap of the raw data. **a**, Differentially accessible peaks identified by EpiScanpy on the raw data, **b**, Differentially accessible peaks identified by EpiScanpy on the data enhanced by scOpen and **c**, Cell type-specific peaks identified by scCASE.

Fig. R28. SNPsea enrichment analysis for the scCASE-identified cell type-specific peaks and accessible peaks identified by EpiScanpy on raw data and data enhanced by baseline methods. **a**, HSC-specific peaks. **b**, GMP-specific peaks.

In summary, cell type-specific peaks identified by scCASE showed relatively better enrichment results than those identified by EpiScanpy in the raw data or data enhanced by baseline methods, demonstrating the biological significance of the cell type-specific peaks identified by scCASE. In the revised manuscript, we have added results of biological analysis in Section “scCASE intuitively reveals cell type-specific biological insights” and Supplementary Text S4. We once again thank the reviewer for this valuable comment.

Fig. R29. Heritability enrichment estimates for the cell type-specific peaks identified by scCASE and accessible peaks identified by EpiScanpy on raw data and data enhanced by baseline methods.

COMMENT 5

5. As a manuscript that mainly contributes computational tools and analyses, it’s better to provide the immediate data and source code for readers to reproduce the results presented in the manuscript.

RESPONSE TO COMMENT 5

We appreciate the reviewer for this valuable suggestion. In this revision, we have followed the reviewer's advice by publishing all the code and immediate data used in the manuscript on Zenodo¹¹, ensuring that readers can reproduce the results presented in the article. Moreover,

we have modified the overall functions of scCASE and facilitated its step-by-step application. We have also diligently revised the tutorial, providing a more detailed description of each function to offer users a more thorough understanding of scCASE. Furthermore, the integration with EpiScanpy facilitates better alignment of scCASE with widely used pipelines, enhancing community collaboration.

References

1. van Dijk, D. et al. Recovering Gene Interactions from Single-Cell Data Using Data Diffusion. *Cell* **174**, 716-729 e727 (2018).
2. Batson, J., Royer, L. & Webber, J. Molecular cross-validation for single-cell RNA-seq. *bioRxiv*, 786269 (2019).
3. Tran, D., Tran, B., Nguyen, H. & Nguyen, T. A novel method for single-cell data imputation using subspace regression. *Sci. Rep.* **12**, 2697 (2022).
4. Tjärnberg, A. et al. Optimal tuning of weighted kNN-and diffusion-based methods for denoising single cell genomics data. *PLoS Comput. Biol.* **17**, e1008569 (2021).
5. Liu, J., Pan, Y., Ruan, Z. & Guo, J. SCDD: a novel single-cell RNA-seq imputation method with diffusion and denoising. *Brief. Bioinformatics* **23**, bbac398 (2022).
6. Cusanovich, D.A. et al. A Single-Cell Atlas of In Vivo Mammalian Chromatin Accessibility. *Cell* **174**, 1309-1324 e1318 (2018).
7. Li, R. & Quon, G. scBFA: modeling detection patterns to mitigate technical noise in large-scale single-cell genomics data. *Genome Biol.* **20**, 193 (2019).
8. Xiong, L. et al. SCALE method for single-cell ATAC-seq analysis via latent feature extraction. *Nat. Commun.* **10**, 4576 (2019).
9. Yuan, H. & Kelley, D.R. scBasset: sequence-based modeling of single-cell ATAC-seq using convolutional neural networks. *Nat. Methods* **19**, 1088-1096 (2022).
10. Li, Z. et al. Chromatin-accessibility estimation from single-cell ATAC-seq data with scOpen. *Nat. Commun.* **12**, 6386 (2021).
11. Tang, S. et al. Accurate and interpretable enhancement for single-cell chromatin accessibility sequencing data with scCASE. *Zenodo*. <https://zenodo.org/records/8382876> (2024).
12. Chen, S. et al. RA3 is a reference-guided approach for epigenetic characterization of single cells. *Nat. Commun.* **12**, 2177 (2021).
13. Ma, W., Su, K. & Wu, H. Evaluation of some aspects in supervised cell type identification for single-cell RNA-seq: classifier, feature selection, and reference construction. *Genome Biol.* **22**, 1-23 (2021).

14. Chen, S., Wang, R., Long, W. & Jiang, R. ASTER: accurately estimating the number of cell types in single-cell chromatin accessibility data. *Bioinformatics* **39** (2023).
15. Granja, J.M. et al. ArchR is a scalable software package for integrative single-cell chromatin accessibility analysis. *Nat. Genet.* **53**, 403-411 (2021).
16. Luecken, M.D. et al. Benchmarking atlas-level data integration in single-cell genomics. *Nat. Methods* **19**, 41-50 (2022).
17. Chen, H. et al. Assessment of computational methods for the analysis of single-cell ATAC-seq data. *Genome Biol.* **20**, 241 (2019).
18. Chen, X. et al. Cell type annotation of single-cell chromatin accessibility data via supervised Bayesian embedding. *Nat. Mach. Intell.* **4**, 116-126 (2022).
19. Muto, Y. et al. Single cell transcriptional and chromatin accessibility profiling redefine cellular heterogeneity in the adult human kidney. *Nat. Commun.* **12**, 2190 (2021).
20. Hao, Y. et al. Integrated analysis of multimodal single-cell data. *Cell* **184**, 3573-3587 e3529 (2021).
21. Xiong, L. et al. Online single-cell data integration through projecting heterogeneous datasets into a common cell-embedding space. *Nat. Commun.* **13**, 6118 (2022).
22. Cao, Z.J. & Gao, G. Multi-omics single-cell data integration and regulatory inference with graph-linked embedding. *Nat. Biotechnol.* **40**, 1458-1466 (2022).
23. King, K.Y. & Goodell, M.A. Inflammatory modulation of HSCs: viewing the HSC as a foundation for the immune response. *Nat. Rev. Immunol.* **11**, 685-692 (2011).
24. Rodrigues, N.P. et al. GATA-2 regulates granulocyte-macrophage progenitor cell function. *Blood* **112**, 4862-4873 (2008).
25. Buenrostro, J.D. et al. Integrated Single-Cell Analysis Maps the Continuous Regulatory Landscape of Human Hematopoietic Differentiation. *Cell* **173**, 1535-1548 e1516 (2018).
26. Buttner, M., Miao, Z., Wolf, F.A., Teichmann, S.A. & Theis, F.J. A test metric for assessing single-cell RNA-seq batch correction. *Nat. Methods* **16**, 43-49 (2019).
27. Danese, A. et al. EpiScanpy: integrated single-cell epigenomic analysis. *Nat. Commun.* **12**, 5228 (2021).
28. Slowikowski, K., Hu, X. & Raychaudhuri, S. SNPsea: an algorithm to identify cell types, tissues and pathways affected by risk loci. *Bioinformatics* **30**, 2496-2497 (2014).

29. Su, A.I. et al. A gene atlas of the mouse and human protein-encoding transcriptomes. *Proc. Natl. Acad. Sci. U.S.A.* **101**, 6062-6067 (2004).
30. Lawrence, M. et al. Software for computing and annotating genomic ranges. *PLoS Comput. Biol.* **9**, e1003118 (2013).
31. Finucane, H.K. et al. Partitioning heritability by functional annotation using genome-wide association summary statistics. *Nat. Genet.* **47**, 1228-1235 (2015).
32. Zamanighomi, M. et al. Unsupervised clustering and epigenetic classification of single cells. *Nat. Commun.* **9**, 2410 (2018).
33. Schep, A.N., Wu, B., Buenrostro, J.D. & Greenleaf, W.J. chromVAR: inferring transcription-factor-associated accessibility from single-cell epigenomic data. *Nat. Methods* **14**, 975-978 (2017).
34. Sandelin, A., Alkema, W., Engström, P., Wasserman, W.W. & Lenhard, B. JASPAR: an open-access database for eukaryotic transcription factor binding profiles. *Nucleic Acids Res.* **32**, D91-D94 (2004).
35. Akhtyamov, P., Shaheen, L., Raevskiy, M., Stupnikov, A. & Medvedeva, Y.A. scATAC-seq preprocessing and imputation evaluation system for visualization, clustering and digital footprinting. *Brief. Bioinformatics* **25** (2023).
36. Liu, Y., Zhang, J., Wang, S., Zeng, X. & Zhang, W. Are dropout imputation methods for scRNA-seq effective for scATAC-seq data? *Brief. Bioinformatics* **23**, bbab442 (2022).
37. Tanigawa, Y., Dyer, E.S. & Bejerano, G. WhichTF is functionally important in your open chromatin data? *PLoS Comput. Biol.* **18**, e1010378 (2022).
38. Lin, X. & Boutros, P.C. Optimization and expansion of non-negative matrix factorization. *BMC Bioinform.* **21**, 7 (2020).
39. Chen, L., Xu, J. & Li, S.C. DeepMF: Deciphering the latent patterns in omics profiles with a deep learning method. *BMC Bioinform.* **20**, 1-13 (2019).
40. Wan, X., Zhang, B., Zou, G. & Chang, F. Sparse data recommendation by fusing continuous imputation denoising autoencoder and neural matrix factorization. *Appl. Sci.* **9**, 54 (2018).
41. Dhont, M., Tsiorkova, E. & González-Deleito, N. Deriving spatio-temporal trajectory fingerprints from mobility data using non-negative matrix factorisation. In *2021 International Conference on Data Mining Workshops* 750-759 (2021).

Reviewer #1 (Remarks to the Author):

The revised manuscript is much more comprehensive. The additional experiments on overmoothing, sequencing depth and scalability shows the robustness of scCASE. The improved implementation to address sequencing-depth correction, scalability and batch correction makes it a useful tool to the single cell community. The authors addressed all of my concerns.

Reviewer #1 (Remarks on code availability):

They provided a README with installation instructions as well as tutorial examples for the main functionality of scCASE.
Source code for reproducible results are included in Zenodo.

Reviewer #2 (Remarks to the Author):

The authors have largely addressed my concerns and have submitted an impressive rebuttal. I have nothing else to add that would improve the manuscript.

Reviewer #3 (Remarks to the Author):

The authors have addressed all my concerns and the revised manuscript is acceptable to me.